



# Field comparison of load-based wind turbine wake tracking with a scanning lidar reference

David Onnen[1,2], Gunner Chr. Larsen[3], Alan W. H. Lio[3], Paul Hulsman[1,2], Martin Kühn[1,2], and Vlaho Petrović[1,2]

[1]Carl von Ossietzky Universität Oldenburg, School of Mathematics and Science, Institute of Physics
[2]ForWind - Center for Wind Energy Research, Küpkersweg 70, 26129 Oldenburg, Germany
[3]Technical University of Denmark, Department of Wind and Energy Systems, Denmark

**Correspondence:** David Onnen (david.onnen@uol.de)

**Abstract.** Wind farm control concepts require awareness and observation methods of the inner-farm flow field. The relative location of the wake, to which a downstream turbine is exposed, is of high interest. It can be used as feedback to support closed-loop wake-steering control, ultimately leading to higher power extraction and fatigue load reduction. With increasing fidelity, not only time-averaged wakes but also instantaneous wake conditions, subject to meandering and wind direction changes, are considered within a controller. This paper presents a quantitative field comparison of two independently applied wake centre estimation methods: a scanning lidar and an Extended Kalman Filter (EKF) based on the rotor loads of the waked turbine. No ground truth is available in the field environment, therefore the methodology accounts for the fact that two uncertain estimates are compared. The lidar estimates, with a derived uncertainty in the order of $0.05$ rotor diameters $D$, can be used as a suitably precise reference to draw conclusions regarding the load-based EKF. The EKF uses Coleman-transformed blade root bending moments, linked to the wake centre position via an analytical model with a low number of tuning parameters. The model can easily be trained with aeroelastic simulations including the Dynamic Wake Meandering model. The formulation adds robustness to the tracking and allows to determine the confidence in the wake position estimate, which can be used for wake impingement detection or for a wake-steering controller to judge whether a yaw manoeuvre is adequate. The results indicate agreement of the methods with root-mean-square errors of $0.2\,D$ for low and moderate turbulence intensity, and $0.3\,D$ for turbulence intensities above 12%. The paper focuses on wake position estimation but also outlines a methodology, how wind farm models or wind field reconstruction techniques can be validated with complementary lidar data.





# 1 Introduction

Wind farm flow control allows to partly compensate wake-induced power losses or load increases. Either wake steering, static induction control, or wake mixing strategies are employed to that purpose (Meyers et al., 2022). So far, mostly open-loop

approaches are considered for wake steering, namely misaligning or dynamically actuating the upstream turbine(s) without considering feedback of the wake-exposed turbines (see e.g. Fleming et al. (2017); Doekemeijer et al. (2021)). Here, the yaw controller relies on engineering models regarding the wake trajectory it tries to aim for. While robust formulations can account for wind direction variability (Rott et al., 2018; Simley et al., 2020), optimal wake deflection cannot be guaranteed, since outer influences and wake dynamics can hardly be encountered. The wake trajectory is impacted by atmospheric stability and further

subject to the meandering motion (Larsen et al., 2015; Sengers et al., 2023).

The consequent next step is to close the loop by providing suitable feedback signals to a wind farm controller. Meyers et al. (2022) explicitly mention the need for state estimation on wind farm level, i.e. for the awareness of the flow conditions within the farm. Standard SCADA data and basic instrumentation of modern wind turbines, e.g. strain gauges for blade root bending moments, allow to use the rotor as a sensor. Rotor effective measurements such as power, torque and collective blade loads

provide observability towards rotor effective wind speeds (Soltani et al. (2013); Bottasso et al. (2018); Lio et al. (2023); ?). This can either be used as direct feedback or to tune an analytical flow model as shown by Doekemeijer and van Wingerden (2020) and Becker et al. (2022). Yet, the observability is limited, as shown e.g. in Doekemeijer and van Wingerden (2020), where the estimator can hardly distinguish which half of the rotor is exposed to a partial wake, especially under uncertain wind direction information. At non-uniform inflow, the spatial observability can be increased by encountering for rotor imbalances

resulting from shear, yaw misalignment or wake impingement Bertelè et al. (2017).

Ultimately relevant for wake-steering control is the wake position within the wind farm, i.e. the feature the controller aims to manipulate. Existing methods for the wake position estimation are either based on wind turbine rotor loads or on Light Detection and Ranging (lidar) measurements. The load-based approaches described by Bottasso et al. (2018) and Schreiber et al. (2020) aim at qualitative impingement detection and include a field validation. Time-averaged position tracking is shown

by Cacciola et al. (2016) in aeroelastic simulations and by Schreiber et al. (2016) in a wind tunnel. Yet, the dynamics caused by wind direction changes and wake meandering are not taken into account here. The lidar-based approaches include the wake deficit shape to the estimation (Raach et al., 2017; Lio et al., 2021) and can also account for the dynamic meandering, as shown in an EKF formulation for a four fixed beam lidar by Lio et al. (2021). Adaptations of the EKF formulation, using blade loads but also taking the meandering dynamics into account, are shown by Dong et al. (2021) and Onnen et al. (2022). Yet, these

methods are only tested in simulation environments, where the wake position is known or can be easily probed in case of large-eddy simulations (LES), as shown by Coudou et al. (2018). The research gap can be concluded as follows: Existing work for load-based wake tracking lacks either a consideration of wake dynamics and time resolution, a field validation, or (in case of a field validation) an independent reference to compare with.

The objective of this work is to fill the gap by performing direct estimation of the instantaneous wake centre position in a

field experiment with two utility-scale wind turbines. The load-based estimate is compared to the wake position probed with





a scanning lidar, which serves as an independent reference. To that purpose, the uncertainty of the lidar estimate is quantified using analytic error propagation following the GUM (Guide to the expression of uncertainty in measurement; JCGM, 2020). The lidar data processing orients at existing work of Trujillo et al. (2011), Machefaux et al. (2015) and Bromm et al. (2018), isolating the quasi-instantaneous wake deficit in a moving frame of reference.

The remainder of the paper is structured as follows: In section 2 the methodology is described, starting with the field setup, followed by the load-based EKF and the lidar-related data processing, including an uncertainty consideration. In section 3 the results are presented. First, the experimental conditions are characterised, then the wake position estimates are compared. In section 4, the findings are discussed, ranged and compared with literature. Concluding remarks are given in section 5.

## 2 Methodology

### 2.1 Field experiment

The wind farm used in this work consists of two Eno126 turbines, built by Eno Energy Systems GmbH near the village Kirch Mulsow near Rostock, Germany close to the baltic sea. The surrounding nature of the test site has agricultural vegetation, with patches of trees and bushes between the fields. The measurements used for this paper are from February and March 2021. Further investigations of the experiments at this site are reported in (Hulsman et al., 2022; Sengers et al., 2023; **?**). The turbines are spaced by 2.7 rotor diameters along south-westerly direction (compare Figure 1), which is also the prevailing wind direction for this site. For brevity, the turbines are called WT1 and WT2 in the following, with WT1 located in the south-west, thus mostly being the upstream turbine. Each turbine has a rotor diameter $D$ of 126 m and a rated power of 3.5 MW. The hub heights are 117 m and 137 m, for WT1 and WT2, respectively. In addition to standard operational signals, both turbines are equipped to measure blade root bending moments in flapwise and edgewise directions with strain gauges and fiber-optical sensors. This paper uses the fiber-optical sensors by Polytech Wind Power Technology Germany GmbH (formerly Fos4X GmbH). Both turbines' nacelle yaw orientation is tracked with interconnected differential Global Navigation Satellite System devices (GNSS; Trimble type 3 Zephyr mode, three antennas on WT1 and two on WT2). The increased accuracy of the yaw angle probing in comparison to the inbuilt yaw encoders is relevant for the post-processing of the lidar measurements, as recommended by Bromm et al. (2018). The rotor azimuth angle information of WT2 was not available, thus the angle was reconstructed from the the gravity-dominated edgewise blade loads as shown in Appendix A.

A met mast is located $2.6\,D$ north of WT1 (see Figure 1). The wind speed and direction are probed with cup anemometers and vanes (Thies Clima type 4.3352.00.400 and type 4.3151.00.212, respectively) at $z_1 = 54$ m and $z_2 = 112$ m. The wind shear exponent $\alpha$ is fitted according to the power law:

$$\alpha = \frac{\log(u_2/u_1)}{\log(z_2/z_1)}\,, \tag{1}$$

where $u_i, i \in 1, 2$ are the wind speed and $z_i, i \in 1, 2$ are the height of the wind cup anemometers. All so far mentioned measurements are stored at 50Hz.





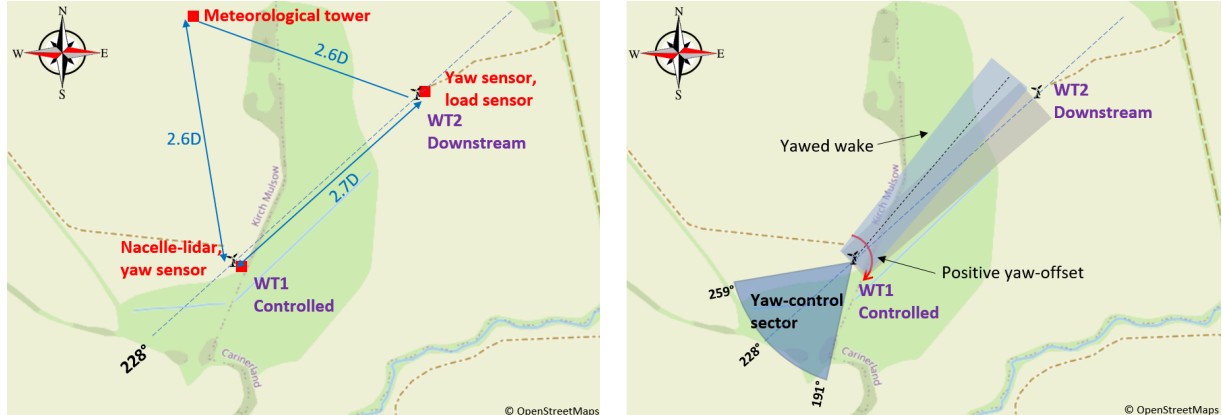

**Figure 1.** Wind farm layout at Kirch Mulsow test site; adapted from Hulsman et al. (2022). © OpenStreetMap contributors 2024. Distributed under the Open Data Commons Open Database License (ODbL) v1.0.

A pulsed scanning lidar (Leosphere WindCube 200S) is installed on the nacelle of WT1, facing in downstream direction. Within the wind direction sector under investigation, the lidar performs horizontal trajectories (single Plan Position Indicator - PPI). The scanned sector covers a range of $120°$ with a scanning speed of $2°\mathrm{s}^{-1}$ and range gates between $50\,\mathrm{m}$ and $1630\,\mathrm{m}$.

The coordinate systems involved in the post processing and further details regarding the lidar trajectory are described in section 2.3.1.

Within the wind direction interval $[191°, 259°]$, active wake steering control is tested. At intervals of $30\,\mathrm{min}$ the controller of WT1 toggles between greedy and intentional yaw misalignment. The yaw update frequency is at $30\mathrm{s}$ and the misalignment is realised via manipulation of the nacelle vane signal. The assessment of the wake steering controller is not the focus of this

paper, yet it is important to regard its role when discussing the wake constellations.

## 2.2 Load-based wake tracking

In this section, the methods used for the load-based wake tracking algorithm and usage of training data are described. Core of the tracking algorithm is an Extended Kalman Filter (EKF), which links the load measurements from a wake-exposed wind turbine with the physical knowledge about the wake behaviour. An EKF incorporates nonlinear state- and measurement

transition functions via local linearisation around the current state estimate. The interaction between the individual aspects of the load-based wake tracking problem is shown in the overview chart in Figure 2. The EKF and its sub-components are described in the following subsections. Note, that the estimation task is here formulated for the general, 2-dimensional case, so considering the horizontal and vertical wake position. Due to the measurement setup and the single PPI scans of the lidar, only a comparison of the horizontal component is possible. The horizontal wake position is also in focus in the context of

wake-steering control.





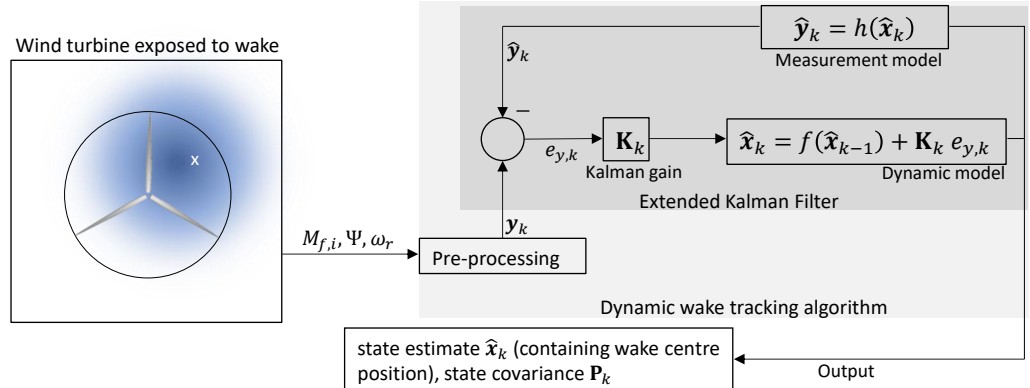

**Figure 2.** Block diagram of the dynamic wake tracking algorithm

### 2.2.1 General EKF setup

A discrete EKF is implemented, where $k$ denotes the time index, $\hat{(\cdot)}$ an estimate, $\boldsymbol{x}_k \in \mathbb{R}^{N_x}$ the state vector and $\boldsymbol{y}_k \in \mathbb{R}^{N_y}$ the measurement vector, with dimensions $N_x = 4$ and $N_y = 3$. The state vector contains the wake positions $(y_\mathrm{w}, z_\mathrm{w})$ as well their first derivatives with time $(v_c, w_c)$. The measurement vector $\boldsymbol{y}_k$ contains the Coleman-transformed non-rotating rotor loads, further described in subsection 2.2.3.

$$\boldsymbol{x}_k = [y_\mathrm{w}, z_\mathrm{w}, v_c, w_c]^T, \qquad \boldsymbol{y}_k = [M_\mathrm{yaw}, M_\mathrm{tilt}, M_\mathrm{col}]^T \tag{2}$$

The model $\boldsymbol{x}_{k+1} = f(\boldsymbol{x}_k, \boldsymbol{n}_{x,k})$ describes the state transition, and the measurement model $\boldsymbol{y}_k = h(\boldsymbol{x}_k, \boldsymbol{n}_{y,k})$ describes the static mapping between the system state and measurements, where $\boldsymbol{n}_{x,k} \in \mathbb{R}^{N_x}$ and $\boldsymbol{n}_{y,k} \in \mathbb{R}^{N_y}$ represent white noise acting on the state and output equation, respectively, with zero mean and covariance matrices $\mathbf{Q}$ and $\mathbf{R}$. The state covariance matrix is denoted $\mathbf{P}_k$. It is initialized as $\mathbf{P}_{k=0} = \mathbf{Q}$. The local linearisations of the state transition model and the measurement model around a current state are denoted $\mathbf{F}_k$ and $\mathbf{H}_k$, respectively. An 'a priori' value is denoted $(\cdot)^-$. The EKF algorithm consists of the following steps.

Prediction Step:

$$\hat{\boldsymbol{x}}_k^- = f(\hat{\boldsymbol{x}}_{k-1}, 0) \tag{3}$$

$$\mathbf{P}_k^- = \mathbf{F}_k \mathbf{P}_{k-1} \mathbf{F}_k^T + \mathbf{Q} \qquad \text{with} \quad \mathbf{F}_k = \frac{\partial f(\boldsymbol{x}_{k-1}, 0)}{\partial \boldsymbol{x}} \tag{4}$$

Measurement Update Step:

$$\mathbf{K}_k = \mathbf{P}_k^- \mathbf{H}_k^T (\mathbf{H}_k \mathbf{P}_k^- \mathbf{H}_k^T + \mathbf{R})^{-1} \text{ with } \mathbf{H}_k = \frac{\partial h(\hat{\boldsymbol{x}}_k^-, 0)}{\partial \boldsymbol{x}} \tag{5}$$

$$\hat{\boldsymbol{x}}_k = \hat{\boldsymbol{x}}_k^- + \mathbf{K}_k \underbrace{(\boldsymbol{y}_k - \hat{\boldsymbol{y}}_k)}_{\text{innovation } e_{y,k}} \qquad \text{with} \quad \hat{\boldsymbol{y}}_k = h(\hat{\boldsymbol{x}}_k^-, 0) \tag{6}$$

$$\mathbf{P}_k = (\mathbf{I} - \mathbf{K}_k \mathbf{H}_k) \mathbf{P}_k^- \tag{7}$$



### 2.2.2 Dynamic model

The dynamic model describes how the system state evolves over time. In this study, the model should capture how the wake
centre position changes over time. Depending on the atmospheric conditions and the wind farm control strategy, the wake
trajectory is subject to various dynamic influences. Time scales of wind direction changes, wake-steering control and wake me-
andering need to be incorporated by the dynamic model of the EKF, while time scales corresponding to small-scale turbulence
with no expressiveness towards the wake position need to be rejected.

For the inner-farm effect of wind direction variability, Simley et al. (2020) suggest a distinction between *low- and high-*
*frequency wind direction*. The high-frequency share refers to oscillatory point-measurements (e.g. a nacelle vane) at hub height
of a wind turbine, while the low-frequency share describes the dominant mean wind direction across the wind farm. Using a
combination of field measurements and LES, Simley et al. (2020) identify the boundary between high- and low-frequency
wind direction at 0.0037 Hz, for a scenario at 8 $\mathrm{ms}^{-1}$ ambient wind speed and wind turbines of 126 m rotor diameter
(NREL 5MW). Rott et al. (2018) suggest to regard a time window of 5 minutes ($\widehat{=}$ 0.0033 Hz), which is very similar. Us-
ing the same non-dimensional type of expression as in (Larsen et al., 2008; Lio et al., 2021), this frequency can be expressed as
$f_{c,WD} \approx u_\infty/(20D)$, where $u_\infty$ is the ambient undisturbed mean wind speed. Depending on the perspective, a high-frequency
wind direction variation can also be seen as the vector addition of longitudinal and transversal wind speed components, so a
turbulence phenomenon.

Wake meandering in the atmospheric boundary layer is driven by turbulence patterns considerably larger than the wake
deficit scale (Trujillo et al., 2011). Larsen et al. (2008) introduced the DWM model, which translates this split of scales to a
random walk trajectory, where the wake deficit is seen as a passive tracer. The default cut-off frequency of the meandering
motion is defined as $f_c = u_\infty/(2D_\mathrm{w})$, where $D_\mathrm{w}$ is the wake diameter (in near wake applications also the rotor diameter $D$
is a valid choice). Note, that this is the theoretical limit, up to which a wake deficit is regarded as a passive tracer. Lio et al.
(2021) show in a field study with a lidar-based EKF featuring an auto-correlation term of the wake position time history that
the dominant spectral share of the meandering motions can be up to a factor 10 slower.

In conclusion, the frequency range of $\frac{u_\infty}{20D} \leq f \leq \frac{u_\infty}{2D}$ is relevant for meandering. Wake position changes at slower time
scales do not require a higher order model. The meandering time scales are thus incorporated with a first-order expression.
This work uses a cut-off frequency of $f_c = 0.01\,\mathrm{Hz}$. The changes in lateral and vertical wake position are described via the
characteristic velocities $v_c(t)$ and $w_c(t)$, whose change rates are modeled as low pass-filtered white noise:

$$\dot{y}_\mathrm{w}(t) = v_c(t) + n_{x,1}(t) \tag{8a}$$

$$\dot{z}_\mathrm{w}(t) = w_c(t) + n_{x,2}(t) \tag{8b}$$

$$\dot{v}_c(t) = -\omega_c\, v_c(t) + \omega_c n_{x,3}(t) \tag{8c}$$

$$\dot{w}_c(t) = -\omega_c\, w_c(t) + \omega_c n_{x,4}(t) \quad , \tag{8d}$$



where $\omega_c = 2\pi f_c$. The equations are discretized for their implementation in the state transition function $f(\boldsymbol{x}_k, \boldsymbol{n}_{x,k})$. Note that the $\boldsymbol{n}_{x,i}$ represents the $i$-th element of the noise vector $\boldsymbol{n}_x$. Since the noise term enters linearly, they are incorporated in the EKF formulation via the additive noise covariance matrix $\mathbf{Q}$.

### 2.2.3 Measurement model

The measurement model is a mapping from the state to the measurement - in this study a link from the wake centre position to the rotor loads. The model must fulfill certain criteria: It should be computationally inexpensive, such that it can be computed online in each filter iteration. Look-up tables with pre-computed information is preferable here (see e.g. Schreiber et al. (2020); Soltani et al. (2013)). Moreover, $h(\boldsymbol{x}_k, \boldsymbol{n}_{y,k})$ has to be differentiable, such that its local sensitivity to a change in state or input can be determined. Finally, it should be robust and lead to a convergence of the estimate, even if the state at initialization is far off.

In the following, a parameterized model is described, which is then fitted to training data. This training data can either be generated in aeroelastic simulations with enabled DWM model (Larsen et al., 2008) or directly from operational data, if a reference wake position is available. The measurement vector $\boldsymbol{y}_k$ contains the Coleman transformed, non-rotating flapwise blade root bending moments according to Equation 9. The time index $k$ is omitted from the notation for better readability.

$$\boldsymbol{y} = \begin{bmatrix} M_{\text{yaw}} \\ M_{\text{tilt}} \\ M_{\text{col}} \end{bmatrix} = \frac{2}{3} \begin{bmatrix} \sin(\Psi) & \sin(\Psi + \frac{2\pi}{3}) & \sin(\Psi + \frac{4\pi}{3}) \\ \cos(\Psi) & \cos(\Psi + \frac{2\pi}{3}) & \cos(\Psi + \frac{4\pi}{3}) \\ 1/2 & 1/2 & 1/2 \end{bmatrix} \begin{bmatrix} M_{\text{f},1} \\ M_{\text{f},2} \\ M_{\text{f},3} \end{bmatrix} + \boldsymbol{n}_y \, , \tag{9}$$

where $\Psi$ denotes the rotor azimuth position and $M_{f,i}$ denotes the $i^{st}$ blade flapwise blade root bending moment.

The yaw and tilt moment depend on the wake position $(y_{\text{w}}, z_{\text{w}})$ relative to the rotor. Let these be expressed in polar coordinates centered at the hub, where $r_{\text{w}} = \sqrt{y_{\text{w}}^2 + z_{\text{w}}^2}$ is the distance of the hub to the wake centre. The ratio between rotor tilt and yaw moment yields information about the angle $\theta$, the angular position of a wake in the rotor plane, quantified as $\theta = \text{atan}\,(y_{\text{w}}/z_{\text{w}})$, using a four-quadrant inverse tangent. In order to get the absolute magnitude of the wake-induced rotor imbalance, the quantity $\tilde{M}(r_{\text{w}})$ is introduced, as

$$\tilde{M}(r_{\text{w}}) = \sqrt{(M_{\text{yaw}}(r_{\text{w}}, \theta) - b)^2 + (M_{\text{tilt}}(r_{\text{w}}, \theta) - c)^2} \, . \tag{10}$$

A reformulation yields the following compact formulation of the yaw and tilt moments,

$$M_{\text{yaw}}(r, \theta) = \tilde{M}(r_{\text{w}}) \cdot \sin(\theta + d) + b \tag{11a}$$

$$M_{\text{tilt}}(r, \theta) = \tilde{M}(r_{\text{w}}) \cdot \cos(\theta + d) + c \, , \tag{11b}$$

where $b, c$ describe offsets to the moments that do not originate from the wake, such as a moment due to tilt overhang or vertical shear. An optional term for phase delay is denoted as $d$, describing how an inert blade reacts to a change of local wind speed, also known as yaw-tilt coupling. The quantity $\tilde{M}(r_{\text{w}})$ is parameterized as in Equation 12, where $\tilde{M}_{\text{max}}$ and $R_{\text{mix}}$ are




fitting constants.

$$
\tilde{M}(r_{\mathrm{w}}) = \begin{cases} \tilde{M}_{\mathrm{max}} \sin\left(\frac{\pi r_{\mathrm{w}}}{2 R_{mix}}\right) & \text{if } |r_{\mathrm{w}}| < R_{\mathrm{mix}} \\ \tilde{M}_{\mathrm{max}} \exp\left(2\left(\frac{r_{\mathrm{w}}}{R_{\mathrm{mix}}} - 1\right)^2\right) & \text{if } |r_{\mathrm{w}}| \geq R_{\mathrm{mix}} \end{cases} \tag{12}
$$

Although wake tracking based only on rotor imbalance ($M_{\mathrm{yaw}}$, $M_{\mathrm{yaw}}$) was found to be possible, the stability and conver-
gence behaviour can be enhanced by including the collective moment $M_{\mathrm{col}}$ (Onnen et al., 2022). The relation between $r_{\mathrm{w}}$
and $M_{\mathrm{col}}$ is linked to the control strategy of the wind turbine and the rotor effective wind speed (REWS). A larger $r_{\mathrm{w}}$ leads
to a higher REWS, until the wake is so far from the rotor centre that no overlap with the rotor takes place and the REWS
approaches $u_{\infty}$. This means that in the partial load region, $M_{\mathrm{col}}$ is suppressed with more wake overlap, so for decreasing $r_{\mathrm{w}}$.
In the full load region, the blades are pitched to keep the power ($\propto u^3$) constant, which implies that the thrust and the flapwise
moments ($\propto u^2$) decrease with increasing wind speed. The highest loading can be seen at rated wind speed. Consequently,
$M_{\mathrm{col}}$ increases with more wake overlap in the full load region, down to the point when the REWS becomes smaller than the
rated wind speed. $M_{\mathrm{col}}$ is fitted with a Gaussian function of $\sigma = R_{mix}$. The constants $M_{\infty}$ and $M_0$ describe the collective
moment for $r_{\mathrm{w}} \to \infty$ and $r_{\mathrm{w}} = 0$ respectively.

$$
M_{\mathrm{col}}(r) = M_{\infty} - (M_{\infty} - M_0) \exp\left(-\frac{r_{\mathrm{w}}^2}{2 R_{mix}^2}\right) \tag{13}
$$

The characteristic shape of the measurement model as described above is illustrated in Figure 3.

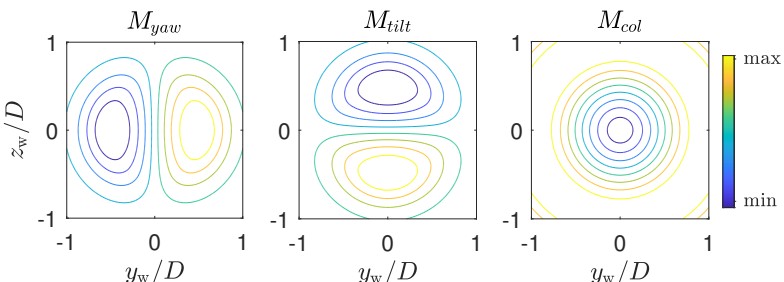

**Figure 3.** Contour plot of the model outputs in dependency of the wake position. Normalized with their respective maximum and minimum
for confidentiality.

The parameters of the measurement model are fitted to a set of training data from aeroelastic simulations within the frame-
work of FASTfarm (Branlard et al., 2022) via a non-linear least-squares regression. The field setup is replicated in the simu-
lations, but the position of WT1 is subsequently shifted laterally from -1.5 $D$ to +1.5 $D$ in steps of 0.5 $D$. The dynamic wake
meandering model (DWM) is enabled, and the curled wake model is chosen (Branlard et al., 2022). As an example, a subset
of the training data at $10\,\mathrm{ms}^{-1}$ ambient wind speed is given in Figure 4, showing the non-dimensionalized yaw moment in
dependency of the wake position $y_{\mathrm{w}}$. Each color refers to one location of WT1. The combination of wake meandering and
different WT1 positions results in a wide range of wake constellations being covered. In case of a constellation with larger


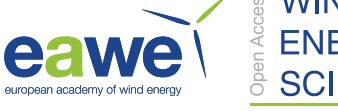

downstream spacing, thus larger meandering amplitudes, even less WT1 positions could be considered for the generation of training data.

The fit parameters depend on the ambient conditions, most prominently on the ambient wind speed. Information of the wake deficit is implicitly contained in the parametric model. Especially in case of large downstream distances, ambient turbulence and atmospheric stability is impacting the wake mixing. In the present case however, the streamwise spacing is too short for the ambient turbulence to show a notable impact on the modeled wake mixing, and thus on the fitting parameters. The impact of shear on the wake deficit is not fully accounted for in the simulation environment, especially in relation to wake-asymmetry

(as discussed later in section 4.1). Thus, it is decided to only create training data in dependency of the ambient wind speed, resulting in a 1-dimensional lookup-table (LUT) of fitting parameters. This requires 63 simulations (7 WT1 positions and 9 wind speeds, 4-12 $\mathrm{ms}^{-1}$), each with a duration of 600s, a TI of 10% and $\alpha = 0.25$. Only one stochastic seed per wind field proved sufficient, since the effect of ambient turbulence is low in comparison to the effect of the wake. Note, that in other cases, e.g. larger streamwise spacing, a higher-dimensional LUT is required, to adequately resolve the impact of turbulent mixing in the far-wake region.

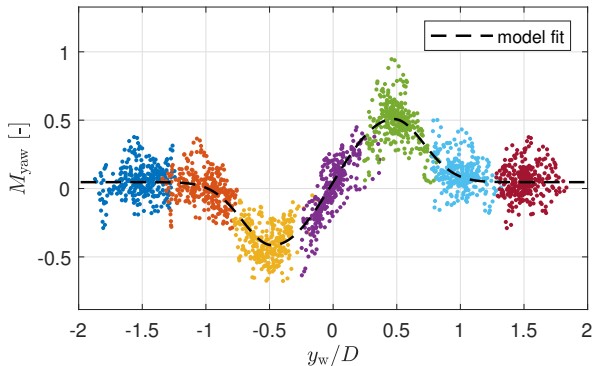

**Figure 4.** Training data and model fit of the yaw moment; each scatter color refers to a prescribed position of WT1. Normalized with the respective maximum and minimum for confidentiality.


In addition to the parameter fitting, the training data allow to seize the order of magnitude of the load variance, linked to turbulence and dynamic events such as load over- and undershoots. This variance is regarded in the noise tuning of the EKF when choosing the entries in the measurement covariance matrix $\mathbf{R}$. The measurement covariance of the yaw moment is increased by a factor of 10 in situations, when the turbine is yawing, to prevent a misinterpretation of the yaw moments

occurring here.

## 2.3  Lidar data processing

This section describes the steps from the initial scanning lidar measurements to a wake position in a WT2-based coordinate system. An uncertainty analysis is included, to show the eligibility of the lidar measurements as a suitably precise reference.





### 2.3.1 Coordinate systems

Different coordinate systems occur in the scope of this work. An overview is shown in Table 1. Ultimately, the lidar-probed wake positions should serve as the reference for the load-based tracking of WT2, thus a WT2-centered coordinate system is targeted. The relations between the coordinate systems are given in Equations (14-16). The lidar performs horizontal single PPI scans. An elevation of $\delta = 1.3°$ is used to account for the average nacelle tilt during operation, determined via the GNSS system on WT1. The azimuth angle $\chi$ covers a range of $120°$ with a scanning speed of $2°\mathrm{s}^{-1}$. The range gate centre is denoted $d_r$ and spans from 50 m to 1630 m in steps of 10 m.

**Table 1.** Coordinate systems

| Coordinate system | Notation | Axes |
|---|---|---|
| Lidar based | $(\cdot)_{Li}$ | spherical, azimuth $\chi$ (positive clockwise), elevation $\delta$, range $d_r$, rotates with WT1 yaw $\gamma_1$ |
| WT1 based | $(\cdot)_{WT1}$ | cartesian, right-handed, $x$ in positive in downstream direction of WT1, $z$: elevation, origin at rotor centre |
| Ground based | $(\cdot)_{GB}$ | cartesian, right-handed, $x$: Easting, $y$: Northing, $z$: elevation, origin at WT1 foundation |
| WT2 based | $(\cdot)_{WT2}$ | cartesian, right-handed, $x$ in positive in downstream direction of WT2, $z$: elevation, origin at rotor centre |


$$\begin{bmatrix} x \\ y \\ z \end{bmatrix}_{WT1} = d_r \begin{bmatrix} \cos(\chi)\cos(\delta) \\ \sin(\chi)\cos(\delta) \\ \sin(\delta) \end{bmatrix}_{Li} \tag{14}$$

$$\begin{bmatrix} x \\ y \\ z \end{bmatrix}_{GB} = \begin{bmatrix} -\sin(\gamma_1) & \cos(\gamma_1) & 0 \\ -\cos(\gamma_1) & -\sin(\gamma_1) & 0 \\ 0 & 0 & 1 \end{bmatrix} \begin{bmatrix} x \\ y \\ z \end{bmatrix}_{WT1} + \begin{bmatrix} 0 \\ 0 \\ h_{WT1} \end{bmatrix} \tag{15}$$

$$\begin{bmatrix} x \\ y \\ z \end{bmatrix}_{WT2} = \begin{bmatrix} -\sin(\gamma_2) & -\cos(\gamma_2) & 0 \\ \cos(\gamma_2) & -\sin(\gamma_2) & 0 \\ 0 & 0 & 1 \end{bmatrix} \begin{bmatrix} x - 252.8\,\mathrm{m} \\ y - 227.6\,\mathrm{m} \\ z \end{bmatrix}_{GB} + \begin{bmatrix} 0 \\ 0 \\ -h_{WT2} \end{bmatrix} \tag{16}$$


### 2.3.2 Wake centre estimation

The horizontal wind speed $u_h$ is the projection of the lidar line-of-sight wind speed $u_{LOS}$ on the wind direction $\Phi$ obtained by the nearby met mast:

$$u_h = \frac{u_{LOS}}{\cos(\chi + \gamma_1 - \Phi)\cos(\delta)} \tag{17}$$





Note that $(\gamma_1 - \Phi)$ expresses the yaw misalignment of WT1. Equation (17) assumes zero lateral and vertical wind speed components $v, w$, which is equivalent to the assumption of identical wind direction at the met mast and the probing position. The impact of this assumption on the uncertainty is discussed in section 2.3.3. The wake position is identified via the horizontal velocity $u_h$ within the upstream area of WT2, defined by $x_{WT2} \in [-110, -90]$ m and $y_{WT2} \in [-200, 200]$ m. The upstream distance is a trade-off between maintaining proximity to WT2 while being less affected by its induction zone (**?**). The subse-

quent wake centre identification is linked to the definition of the wake centre itself, as discussed by Vollmer et al. (2016) and Coudou et al. (2018). A comprehensive overview of different lidar-based tracking methodologies is given by Trujillo (2017). Following Vollmer et al. (2016), a robust approach via the minimum in density of virtual available power is used:

$$y_{\mathrm{w}} = \underset{y_{WT2}}{\arg\max}(p * f_M) \qquad \text{with} \qquad f_M(y_{WT2}) = \begin{cases} -1 & \text{if} \quad |y_{WT2}| \leq \frac{D}{2} \\ 0 & \text{otherwise} \end{cases} \tag{18}$$

where $(.*.)$ denotes a convolution and $f_M$ a square-shaped masking function. The density of available power is defined as

$p(y_{WT2}) = u_h^3$.

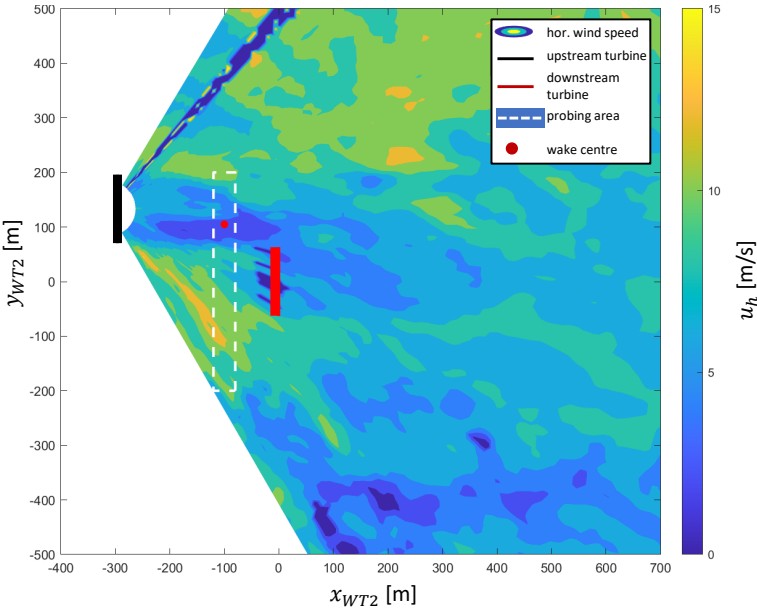

**Figure 5.** Wake centre identification in WT2-based coordinate system

### 2.3.3 Uncertainty estimation

The uncertainty of the wake position $y_{\mathrm{w}}$ is subject to

- the lidar probe position uncertainty

- the uncertainty in the horizontal wind speed at the probe position, when projected from the line-of-sight velocity

- the sensitivity of the wake centre identification method towards the wind speed uncertainty





In principle, an uncertain probe position could further influence the probed wind speed, e.g. when measuring at a different altitude than expected in a sheared or veered flow. This can be corrected for, as shown by Schneemann et al. (2021) for a long-range lidar experiment with range gates of multiple kilometers. In the work presented here, the probe position uncertainty is sufficiently small to neglect the effect of wind speed gradients at the probe position (see later in Figure 6). The probe position is
subject to the measurement uncertainties listed in Table 2. Their propagation through the coordinate transform in Equations (14-16) is formulated by Equation (19), following the GUM standard (JCGM, 2020). The uncertainties are illustrated in Figure 6 for a full alignment case ($\Phi = \gamma_1 = \gamma_2 = 228°$). Note, that the uncertainties depend on the instantaneous constellation.

**Table 2.** Uncertainties in the scope of the lidar data processing; values relate to the 95% confidence interval for normally distributed uncertainties

| Quantity | Variable | Uncertainty |
|---|---|---|
| Lidar elevation | $\delta$ | $\pm 2°$ (impacted by WT1 tilt motion) |
| Lidar azimuth | $\chi$ | $\pm 0.5°$ (see Schneemann et al. (2021)) |
| Range gate centre* | $d_r$ | $2\,\mathrm{m}$ |
| Mean wind direction | $\Phi$ | $\pm 2°$ (see Schneemann et al. (2021); Simley et al. (2020)) |
| WT1 yaw (GNSS based) | $\gamma_1$ | $\pm 0.5°$ |
| WT2 yaw (GNSS based) | $\gamma_2$ | $\pm 0.5°$ |
| LOS wind speed | $u_{LOS}$ | $\pm\, 0.1\ \mathrm{ms}^{-1}$ |

*range gate centre as a result of pulse length and time of travel; the range gate volume is considerably larger

$$\begin{bmatrix} \Delta x \\ \Delta y \\ \Delta z \end{bmatrix}_{WT2} = \sqrt{\left(\frac{\partial \boldsymbol{x}_{WT2}}{\partial \gamma_1}\Delta\gamma_1\right)^2 + \left(\frac{\partial \boldsymbol{x}_{WT2}}{\partial \gamma_2}\Delta\gamma_2\right)^2 + \left(\frac{\partial \boldsymbol{x}_{WT2}}{\partial \delta}\Delta\delta\right)^2 + \left(\frac{\partial \boldsymbol{x}_{WT2}}{\partial \chi}\Delta\chi\right)^2} \tag{19}$$


The uncertainty in the line-of-sight projection of the wind speed $\Delta u_h$ can again be expressed considering the geometry:

$$\Delta u_h = \sqrt{\left(\frac{\partial u_h}{\partial \chi}\Delta\chi\right)^2 + \left(\frac{\partial u_h}{\partial \gamma_1}\Delta\gamma_1\right)^2 + \left(\frac{\partial u_h}{\partial \Phi}\Delta\Phi\right)^2 + \left(\frac{\partial u_h}{\partial \delta}\Delta\delta\right)^2 + \left(\frac{\partial u_h}{\partial u_{LOS}}\Delta u_{LOS}\right)^2} \tag{20}$$

The impact of the wind speed uncertainty on the wake centre identification is investigated. If the wind speed uncertainties were
randomly distributed along $y_{WT2}$, the convolution integral would hardly be affected, since it smoothens on a scale of $1\,D$. However, it is more likely to have a wind speed uncertainty which is correlated along $y_{WT2}$, e.g. as the result of a misaligned lidar beam. This would promote wind speeds at one end of the probing area while suppressing them at the other end. The bias would have a magnitude of $\pm 5\%$ within the wake probing range, as visible in Figure 6d. Figure 7 shows a normalized wake




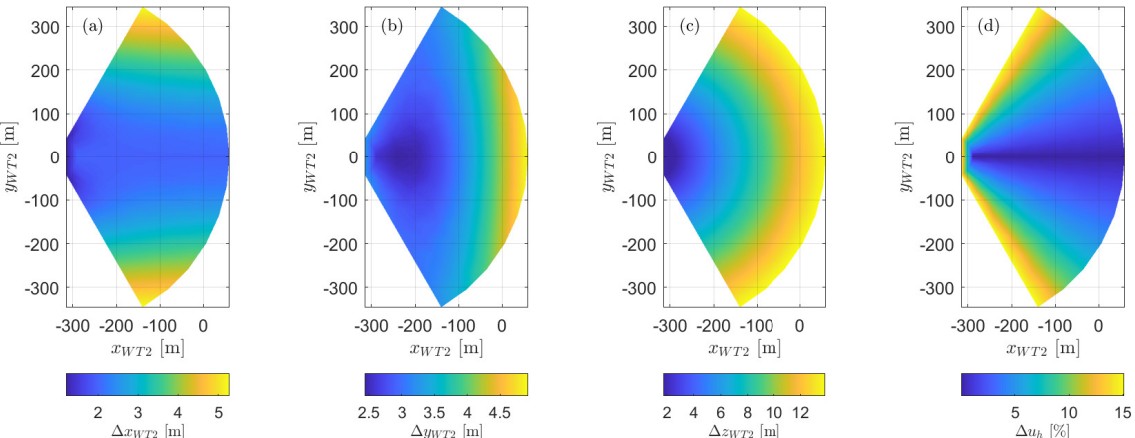

**Figure 6.** Illustration of uncertainty propagation: Probe position uncertainty in $x, y, z$ and horizontal wind speed uncertainty; WT2-based coordinate system

deficit example, which is corrupted by a linear bias of $\pm 5\%$. Applying the convolution method according to Equation (18)

yields a mis-assessment in the order of $\pm 1$ m ( $< 0.01\,D$) for all possible wake positions $y_\mathrm{w}$. Note, that this is no longer a standard uncertainty according to the GUM, since it contains the worst-case assumption of a linear bias. Also note that the mis-assessment of the method depends on the wake deficit. A less pronounced wake deficit would have less impact in comparison to a correlated wind speed uncertainty. Qualitatively, the investigation showed the convolution method to be very robust and hardly affected by the expected range of wind speed uncertainty. The uncertainty of the probe $y$-position and the wake centre

identification uncertainty are subsequently added. The calculation is applied for each full lidar snapshot individually.

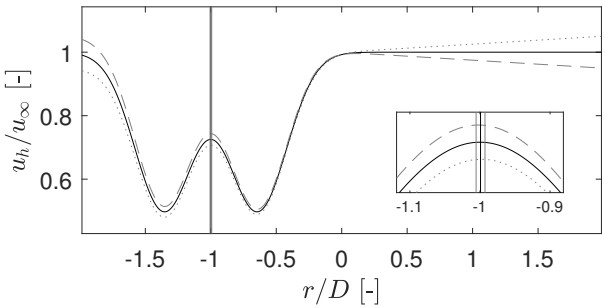

**Figure 7.** Example of convolution method to extract the wake centre position; impact of wind speed uncertainty on wake centre identification: Generic double-Gaussian deficit as found situ is distorted by two exemplary linear biases of $\pm 5\%$, which can be obtained from Figure 6





## 3 Results

### 3.1 Wake condition characterisation

This first part of the result section gives an overview of the wake conditions contained in the test data set. A histogram of the ambient conditions is shown in Figure 8. In total, 1800 one-minute samples (30 hours) of wake constellations are contained. The wind speed distribution does not show the converged shape of a Weibull distribution yet, but the tendency is recognizable. A large share of high shear and low turbulence intensity is on hand, which is an indicator for very stably atmospheric conditions. Note here, that atmospheric stability often follows diurnal cycles, while the data set only contains data from 05:00 to 21:00 UTC, since the wind turbines often operated at a noise reduction mode during night time, which would not have been representative. Also note, that the indicated yaw misalignment does not distinguish between intentional and unintentional yaw misalignment.

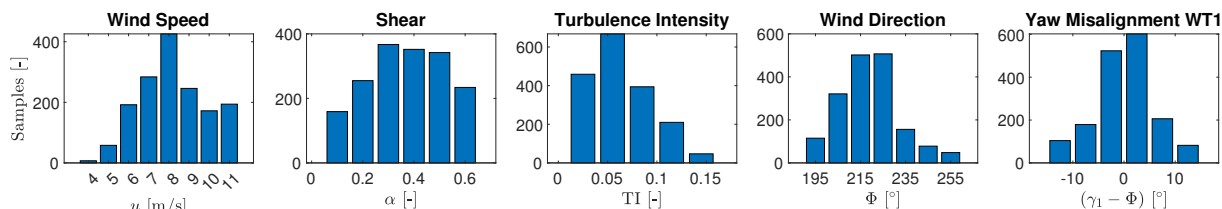

**Figure 8.** Histogram of ambient conditions contained in the investigated data set; all measurements refer the met mast (except for $\gamma_1$, which is probed via GNSS on WT1)

### 3.1.1 Wake position variability

In Figure 9, the wake position $y_\mathrm{w}$, as identified from the lidar scans, is plotted versus the wind direction. It shows higher spreading than suggested by the pure geometry, namely the turbine positions and the assumption of linear wake propagation parallel to the wind direction, indicated in red. The spreading has a magnitude of up to $\approx 40\,\mathrm{m}$ or $0.3\,D$, which is considerably larger than the order of uncertainty connected with the wake centre identification, as discussed in section 2.3.3. The spreading could originate from wake meandering, wind direction changes propagating through the test field, and wake steering control. The impact of the wake steering controller can be estimated by employing the analytical wake deflection model of Jiménez et al. (2009) or Larsen et al. (2020) and the available information of the yaw misalignment of WT1. These models give similar results, but the latter does not require any parameter fitting. Figure 9b shows results for the Jiménez model, which on average successfully encounters the direction and magnitude of wake deflections for the sector in which wake steering control was active $(191° - 259°)$. Note, that toggling between conventional and wake steering control was on hand, thus also many situations with no intentional yaw misalignment are contained in the plot. While the scattering range resulting from the Jiménez model is similar to the observed scattering seen in the lidar data, these scatters are not necessarily concurrent in time. Also, the double-sided deviations of the lidar probed wake positions from the geometry line are not captured.





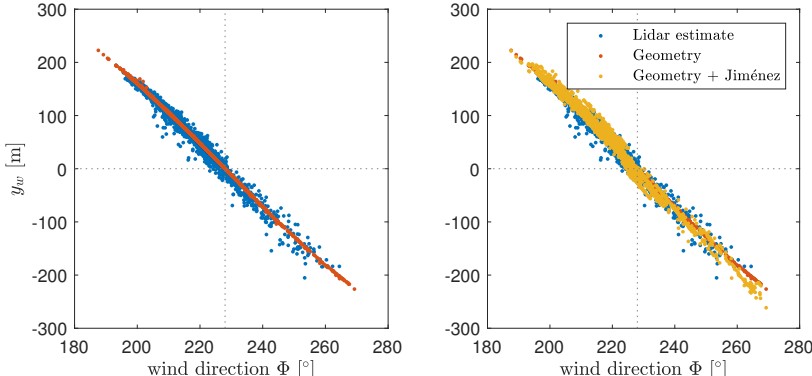

**Figure 9.** Wake centre position $y_\mathrm{w}$ in dependency of wind direction: *Geometry* denotes the pure consideration of farm geometry and linear wake propagation in main wind direction, *Jimenéz* denotes an analytic wake deflection model. Center lines of zero deflection and full turbine alignment (228°) are marked.

### 3.1.2 Wake deficits

Figure 10 shows the wake deficits recorded by the lidar, superimposed within wind speed bins of 0.5 m/s. The instantaneous deficits are aligned along their identified wake centre, thus the horizontal axis in Figure 10 is defined as $r = y_{WT2} - y_\mathrm{w}$ (compare section 2.3.1). Each snapshot is plotted transparent, such that darker areas indicate higher occurrences of similar deficits. Some individual wake deficits differ considerably from the dominant bin average. The wake deficit shows the characteristic double-Gaussian shape of a near wake. Especially for larger wind speeds, a strong asymmetry is observed, pronouncing the wake at negative coordinates $r$ (referring to the right side when facing downstream; compare Figure 5). The asymmetry persist even when filtering the data set for yaw-misalignment situations, which are known from simulations to cause a kidney-shaped curled wake (see e.g. Bartl et al. (2018) and Sengers et al. (2023)). However, an impact is visible when filtering for the power law coefficient $\alpha$, describing the shear profile. Figure 11 indicates that the wake asymmetry is more pronounced at strong shear, connected to atmospheric stable conditions. For low shear coefficients, the wake deficits are rather symmetric. Larger wind speed variations among the deficits as well as in the non-waked area are on hand here, which again is attributed to the atmospheric stability. The role of the wake deficit in this context is further discussed in section 4.

### 3.2 Wake position estimates

This section shows the behaviour of the wake position estimation via load-based EKF and lidar recordings under various ambient conditions. Details are shown in a time series plot and lidar snapshots of the flow field, while the general performance is seized with bar plots of performance metrics applied on the entire data set.

Figure 12 shows a six hour wake position time series on 19th of February, including several lidar recorded snapshots of the instantaneous flow situation in the wind farm. The corresponding ambient conditions as recorded by the nearby metmast are





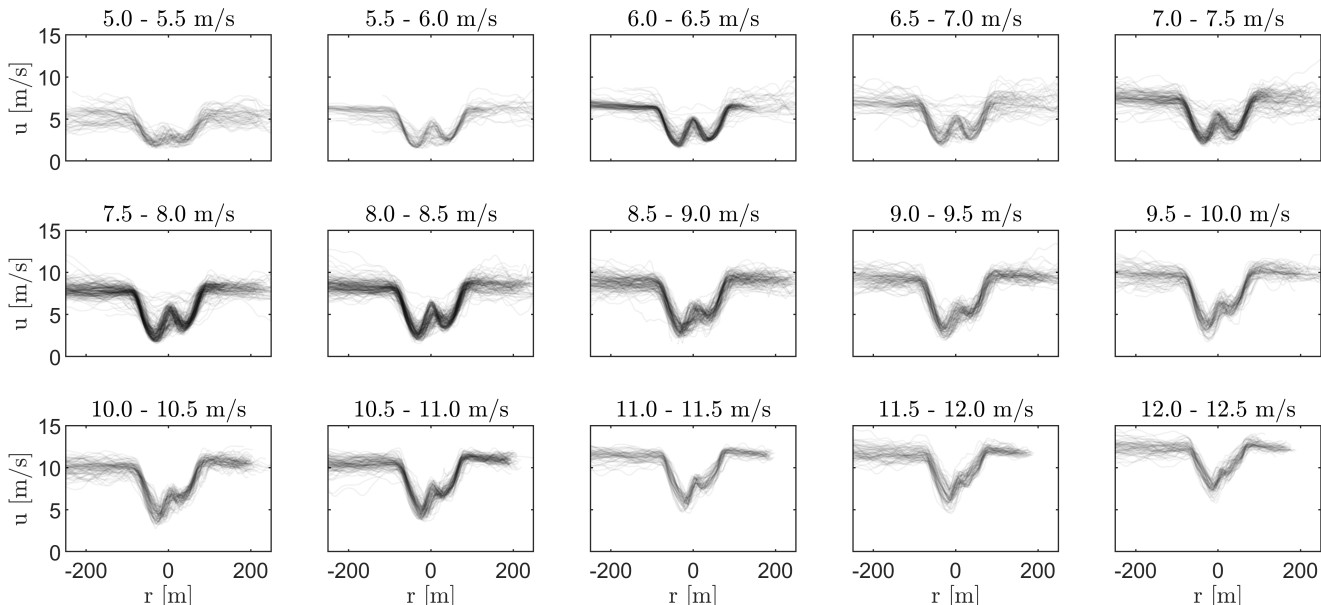

**Figure 10.** Spaghetti plot of the observed wake deficits per individual lidar scan, aligned along their identified wake centre; binned in steps of $0.5\,\mathrm{ms^{-1}}$; plotted transparent to visualize frequent occurrences of similar wake deficits

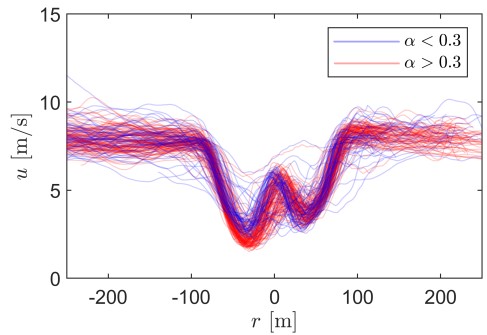

**Figure 11.** Wake deficits within wind speed bin $7.5 - 8\,\mathrm{ms^{-1}}$; colour coded for two ranges of shear profile, defined by power law coefficient $\alpha$

shown in Figure 13. Within the shown time span, the wind direction changes from 250° to 200°, resulting in a full sweep of the
wake across the rotor of WT2 (full alignment is at 228°). Constellations of partial wake, full wake and barely impinging wake
are covered. At the same time, the wind ramps up from 5 to $9\,\mathrm{ms^{-1}}$, and the atmospheric conditions change from unstable to
stable, indicated by high TI, high wind direction variability and low $\alpha$ in the afternoon compared to low TI, low wind direction
variability and high $\alpha$ in the evening hours. The EKF is initialized at $y_{\mathrm{w}} = 0\,$ m and converges to the approximate wake position
within approximately 2 min. Snapshots associated with a variety of conditions - labeled a) to e) - are analyzed in detail.



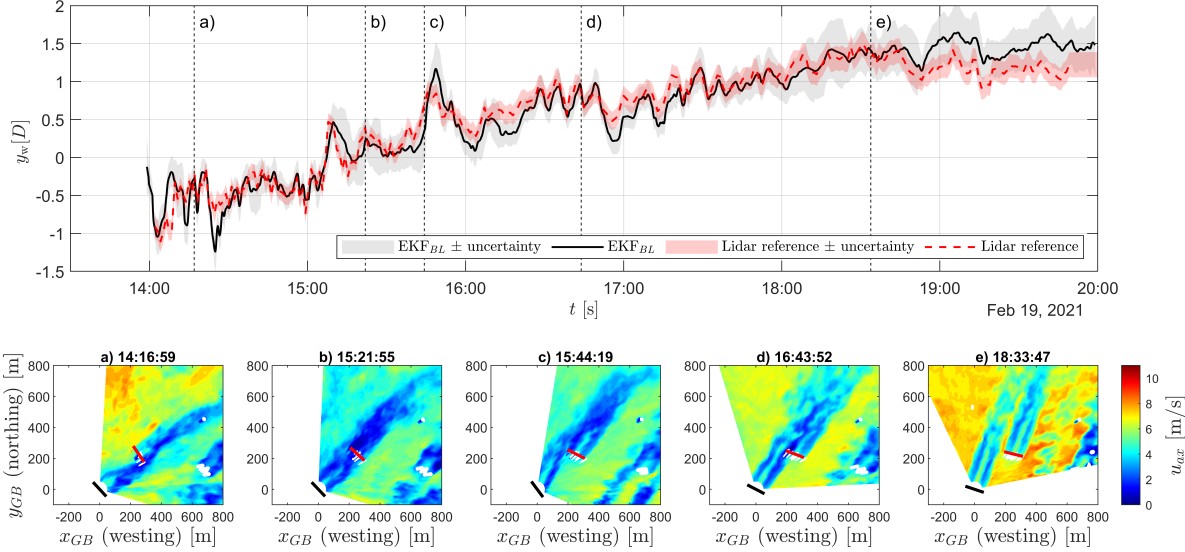

**Figure 12.** Top: Time series of wake position estimate by load-based EKF and lidar; the uncertainty range for both methods is indicated Bottom: Snapshots of the instantaneous flow situation in the wind farm; ground-based coordinates are used; WT1 indicated in black, WT2 in red; the time instances a-e refer to the indications in the time series plot on top

a) Partial wake (at 14:16): A wake constellation at $y_{\mathrm{w}} \approx -D/2$, which agrees with the EKF estimate. The flow dynamics are high at this point in time, which can be seen in the position changes captured by the EKF as well as in the wind field.

b) Full wake (at 15:21): While correctly identified by the EKF, the confidence interval of the EKF is slightly increased here. This is connected to decreased observability, a result of the flat gradient $\left.\frac{\partial M_{col}}{\partial y_{\mathrm{w}}}\right|_{y_{\mathrm{w}} \approx 0}$ used by the local linearisation of the measurement model.

c) Yaw misalignment (from 15:40 to 16:00): A high yaw misalignment ($\approx 15°$) of WT1 is present. The wake steering effect displays with a prominent wake position change, which is also visible in the flow situation of snapshot c. Both the lidar and the EKF capture the steep change in wake position in this time span.

d) Meandering (16:30 and 16:50): The wake position oscillates several times between $0.5\,D$ and $1\,D$. The time scales of these oscillations are around $300\,\mathrm{s}$ (referring to spatial scales of $14\,D$ at $6\,\mathrm{ms}^{-1}$ ambient wind speed, compare section 2.2.2). This is at the higher end of the dynamic range of the EKF, yet close to the transition between what is defined as meandering and as farm-effective wind direction variability.

e ) Barely impinging wake (from 18:00 to 18:40): The wind direction approaches 200° and the average wake position moves from $1\,D$ to $1.5\,D$, which leads to ceasing wake impingement. The loss of observability goes along with increased state covariance, thus a larger confidence interval of the EKF estimate. In case of no wake impingement, the $2\sigma$ confidence is close to $0.5\,D$ for multiple time instances in a row (in case of a single iteration increase, it could also mean a measurement outlier). The EKF position estimate stays approximately at the last known position but cannot be regarded as expressive here.

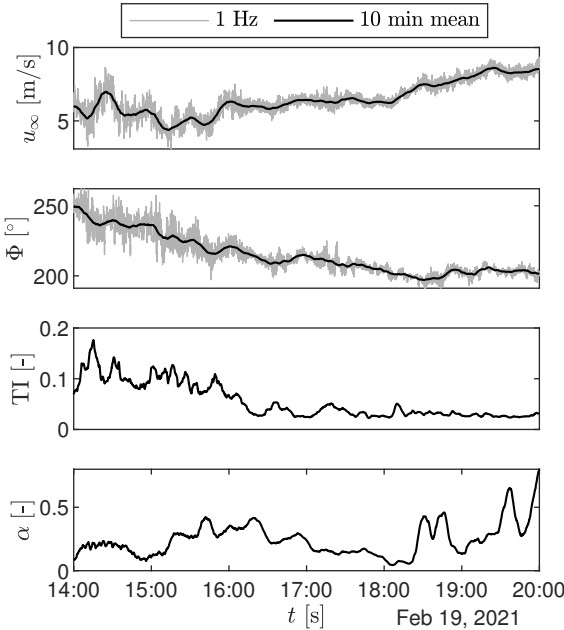

**Figure 13.** Ambient conditions on 19th of February, same time instance as shown in Figure 12; ambient wind speed $u_\infty$ and wind direction $\Phi$ are shown both as 1Hz and 10min average; the TI refers to 10min bins by definition and the same was applied to power law coefficient $\alpha$

The EKF behaviour can further be assessed based on a spectra of the wake position time series, given in Figure 14. The cut-off frequency of the EKF formulation $f_c$ is indicated, which is also close to the band limit of the lidar scanning speed. Within $10^{-3}$ Hz to $10^{-2}$ Hz, the PSD of EKF and lidar estimates is similar and decays with approximately -20 dB/dec. At higher frequencies, the EKF shows a trend of -40 dB/dec, where the additional attenuation is linked to the filter formulation. The filter also contributes to the rejection of changes in wake position faster than $f_c$, which might be suggested by higher-order load variations.

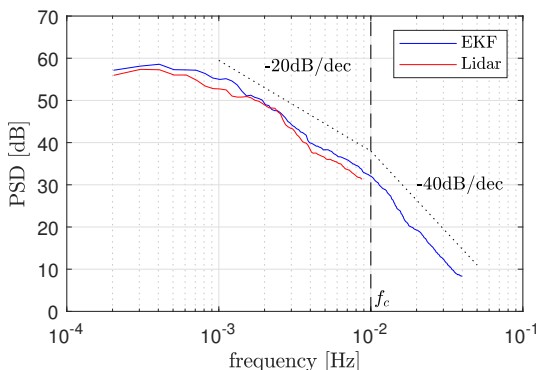

**Figure 14.** Power spectral density (PSD) of the wake position $y_w$, estimated by EKF and by Lidar





The performance of the entire test data set is ranged with performance metrics. The estimates of lidar and EKF are compared with the root-mean-squared-error (RMSE), defined as

$$\text{RMSE} = \sqrt{\frac{1}{N}\sum_{k=1}^{N}(y_{\text{w},k}^{\text{EKF}} - y_{\text{w},k}^{\text{L}})^2} \quad . \tag{21}$$

The RMSE does not capture the uncertainty consideration yet. The additional metric *inRange* is introduced, which denotes whether the estimates are within each others $2\sigma$ uncertainty range. It further accounts for the fact, that no ground truth exists. Instead, two uncertainty-containing signals are compared.

$$inRange = \frac{1}{N}\sum_{k=1}^{N}\Omega_k \qquad \text{with} \quad \Omega_k = \begin{cases} 1 & \left| y_{\text{w},k}^{\text{EKF}} - y_{\text{w},k}^{\text{L}} \right| < 2\sqrt{(\sigma_k^{\text{EKF}})^2 + (\sigma_k^{\text{L}})^2} \\ 0 & \text{else} \end{cases} \tag{22}$$

The results are shown in Figure 15, where binning with respect to the ambient conditions is applied, revealing the dependency on ambient wind speed, shear, TI and WT1 yaw misalignment. The share of data within the respective data bin is shown in Figure 8, to allow assessing the results under consideration of the underlying statistical evidence. E.g., the RMSE of a bin that contains only 3 % of the available data can be considered less expressive than a bin that represents 20 % of the data set. The RMSE is generally around $0.2\,D$ and the *inRange* indicator around 90 %. No clear systematic dependency towards ambient wind speed and shear level is seen. The RMSE varies slightly among the bins, yet the $inRange$ indicator is not notably affected. A trend for the turbulence intensity is visible, namely from $0.2\,D$ RMSE and $inRange$ of 95 % at low TI to $0.3\,D$ RMSE and $inRange$ of 75 % at high TI. The data availability decreases towards higher TI, yet the trend is persistent over all bins and for both metrics. At small yaw misalignments of WT1 the RMSE is lowest. Strong negative yaw misalignments seem to increase the RMSE. Yet this finding is to be treated with care, since the data availability is comparably low here.

## 4 Discussion

### 4.1 Evaluation of the experimental conditions and data processing methods

This section discusses the influence of the site specifications on the results and considers their generalizability.

*Site and wake conditions*

The test site has very close spacing between the turbines, resulting in a near wake with characteristic double-Gaussian deficit shape. The consequence for the estimation task is twofold. On the one hand, the wind speed deficit is very pronounced, so it leaves a considerable footprint on the rotor of a subsequent turbine. On the other hand, a double-Gaussian wake deficit is a more complex structure, thus requiring higher degrees of freedom for its description in comparison to a single-Gaussian (Keane et al., 2016). The scanning lidar can resolve this and even an EKF-based four fixed-beam staring lidar approach as described by Lio et al. (2021) shows sufficient observability. Existing works on estimation using turbine measurements either do not





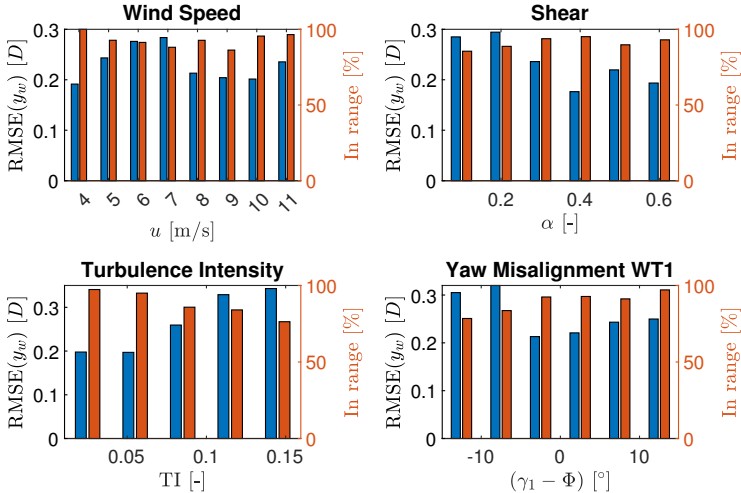

**Figure 15.** RMSE of wake position estimate (EKF vs. lidar), binned with respect to the ambient conditions; the orange bars refer to the right y-axis and represent the *inRange* indicator, so whether the difference between the position estimates is covered by their uncertainty intervals

consider near wake features (Doekemeijer and van Wingerden, 2020; Cacciola et al., 2016), or assume a quasi-steady wake velocity deficit to be known a priori (Dong et al., 2021). The latter is similar to this work, where the wake deficit is implicitly contained in the training data. Even more complexity is added due to the occasional wake asymmetry, reported in the context of Figure 10. As described, no strong correlation of wake asymmetry and WT1 yaw misalignment was found, as it might have been expected with regard to the curled wake phenomena (Bartl et al., 2018; Sengers et al., 2023). Instead, the wake asymmetry

is found to rather co-occur with strong wind shear and increase with ambient wind speed, thus rotational speed. An interaction of wake rotation and the sheared flow is assumed. The rotational component in the wake flow, in opposite direction to the rotor rotation, could cause an 'upwash' of wind speeds from low altitudes on the right side of the rotor (facing downstream, thus negative on the y-axis) and a 'downwash' of wind speeds from higher altitudes on the left side. The direction of wake rotation and the observed orientation of the wake asymmetry would support this explanation. A comparable near wake asymmetry is

reported by Bromm et al. (2018) in a similar field campaign. Another consequence of small downstream distance is a low meandering amplitude (Machefaux et al., 2015). It is expected that the load-based EKF would have been able to capture higher meandering amplitudes, as shown in a wind tunnel experiment with tailored meandering wake conditions (Onnen et al., 2023). In the given field setup, however, a considerable share of the involved wake position dynamics can be accounted on wind direction changes and active yaw control.

*Uncertainty*

The uncertainty consideration for the lidar estimate is deliberately chosen to be mainly based on analytical error propagation rather than on statistical approaches. On the one hand, this choice allows to identify and unravel the impact of individual quantities' contributions to the combined uncertainty of the processed wake position. In this case, the wind speed uncertainty



shows negligible impact when locating a coherent flow structure. The major contributions originate from the propagation of
geometric uncertainties. These can be limited with adequately precise measurement equipment, such as the GNSS encoders for
the nacelle yaw probing used in this setup. On the other hand, the uncertainty is available for every time instance independently,
thus not depending on the data set size, as it would be the case for a statistically derived uncertainty. The lidar estimate generally
has a combined $2\sigma$-uncertainty below $0.06\,D$, which makes it a suitable reference in comparison to the difference between the
lidar- and the load-based method, which is at the order of $0.2\,D$ (RMSE). Trujillo (2017) names $0.05\,D$ as the accuracy of
lidar-based wake position extraction at short downstream distances, which is very similar to this work. The uncertainty of
the EKF estimate is directly taken from its state covariance matrix (Eichstadt et al., 2016). The involved linearisation of the
measurement model is similar to the first-order approximations used for analytic error propagation.

## 4.2 Wake tracking performance

In contrast to a simulation study, a pure performance assessment of one wake tracking methodology is not possible in a field
experiment, since no ground truth exists as reference. Instead, two uncertainty-containing estimates from two different methods
are compared. The wake position estimated with the scanning lidar can be regarded as an attempt to provide a reference value
closer to a virtual ground truth.

*Impact of ambient conditions*
The match between lidar and load-based position estimates shows no clear dependency on the ambient wind speed. Small
variations among the bins could originate from a limited data set size, which might not equal out coinciding instances of
certain wind speeds with e.g. certain turbulence intensities. An indication for a not fully converged data set is the wind speed
histogram in Figure 8, showing that the occurring wind speeds do not fully represent the shape of a Weibull distribution. In a
simulation study, no direct impact of the ambient wind speed on the estimation is reported, as long as both the wake-causing
turbine and the estimating turbine are not operating at the transition of partial to full load range (Onnen et al., 2022).

The observed increase of RMSE with TI is expected and agrees with simulation studies of load-based estimation (Dong et al.,
2021; Onnen et al., 2022) and field results of lidar-based wake estimation (Lio et al., 2021). Higher turbulence intensities affect
both the shape of the instantaneous wake deficit and the dynamics of the wake position. The information contained in the blade
root loads is typically not sufficient to distinguish between both aspects, especially when their characteristic time scales are
overlapping. The definition of the cut-off frequency in the dynamic model of the EKF leads to a rejection of turbulent scales
smaller than the rotor scale. Deviations of the wake deficit shape that persist at scales of multiple rotor diameters could be
misinterpreted as a change in wake position. This holds for the method described in this work, where wake deficit information
is indirectly contained in the training data, as well as for methods that aim to estimate the wake deficit online, yet on slow time
scales (see Lio et al. (2021)). The relation between the instantaneous wake deficit and the wake centre position further impacts
the respective definitions of the wake centre: The convolution with density of available power (as applied on the lidar data,
compare section 2.3.2) always considers the entire wake deficit. In case of non-symmetry it identifies its centre with a shift





towards the more pronounced side of the wake deficit. The load based method, however, solely judges the share of the deficit which overlaps with the estimating turbine.

An impact of the wind shear on of the tracking performance could have been expected, as the asymmetry of the wake deficit shows to be influenced. Yet, the low shears often coincide with high TI, both as features of atmospheric instability. It is not
possible to fully isolate the effects from shear and TI, and although the wake asymmetry due to high shears would lead us to expect a worse tracking performance, this was not observed.

*Comparison to other wake estimation methods*

The comparison to other existing methods considers the respective performance metrics, the test environment (simulations, wind tunnel, or field), and the underlying methods and assumptions. Cacciola et al. (2016) show static inaccuracies of 0.1-
$0.2\,D$ for the determination of the wake centre position at TI$= 5\,\%, 10\,\%$ in aeroelastic simulations. Each position estimate is based on 10 min averaging and a least-squares-fit of rotor-effective horizontal shear with respect to the rotor loads. Bottasso et al. (2018) show detection ratios per location interval (discretized with $0.25\,D$) as a performance measure. The detection method compares the difference in EKF-estimated sector-effective wind speeds with a threshold, which again is subject to scheduling with the ambient conditions. It is also tested in an aeroelastic environment, both in static wake conditions and in a
scenario where a single-Gaussian wake deficit follows a sine trajectory at a frequency of $f \approx u_\infty/(2D)$. The simulations allow for a ground truth reference, but other than that, the detection ratio is similar to the $inRange$ metric used in this paper. Bottasso et al. (2018) show a detection ratio close to $100\,\%$ for static wakes and $5\,\%$ TI, which decreases to approximately $75 - 80\,\%$ at $10\,\%$ TI. This is similar to the results reported in Figure 15. The works also agree that ambient shear decreases the accuracy, while estimation is still possible under moderate yaw misalignment of the tracking turbine. At full wake constellations, the
method of Bottasso et al. (2018) has no observability, because the wake-induced rotor loads are not asymmetric. Here, the comparison between the methods lacks, because they do not use the undisturbed wind speed. But, as also pointed out by the authors, the blind spot at full wake could be avoided when comparing the ambient wind speed with the rotor-effective wind speed, or redundantly with the collective blade loads, as done in this paper. Onnen et al. (2022) test a nearly identical EKF formulation as in this work with aeroelastic simulations using the DWM model and the DTU 10MW turbine. RMSE of $0.05\,D$,
$0.1\,D, 0.2\,D$ is found for turbulence intensities of $5\,\%, 10\,\%, 15\,\%$, respectively. A similar RMSE is shown in another aeroelastic study by Dong et al. (2021) with a similar load-based EKF. In general, the field test shows increased RMSE in comparison to the simulational tests, which likely occurs due to the more uncertain environment. The qualitative tracking ability, as quantitized with the detection ratio or $inRange$ indicator is not notably impacted.

Wind tunnel results with two model turbines of $2\,\mathrm{m}$ diameter are shown by Schreiber et al. (2016). The methodology is
similar to the one of Cacciola et al. (2016), and a time averaging of $1\,\mathrm{min}$ is used, which corresponds to approximately $1\,\mathrm{hour}$ in the field, considering the scaling. Static inaccuracies of $0.1$-$0.2\,D$ are found, in sheared inflow and $8\,\%$ TI. Dynamic wind tunnel tests are shown by Onnen et al. (2023), where a $1.8\,\mathrm{m}$ model turbine is exposed to wake conditions tailored with an active grid. The estimation accuracy is below $0.1\,D$ (RMSE). This is considerable lower than in the field tests shown here, most likely due to the controlled environment, a low ambient TI and no wind direction variability.





To the author's knowledge, the only field test of load-based wake estimation is reported by Schreiber et al. (2020). Qualitative wake impingement detection (left / right / full wake) is successfully shown, where the farm layout and the assumption of wake propagation parallel to the met mast wind direction serve as a reference. The availability of a scanning lidar in this work allows for a quantitative assessment while probing with higher spatial and temporal resolution. Further field experiments with scanning lidar-based wake position identification are reported by Bromm et al. (2018), where a propagated uncertainty of 0.05-0.1 $D$ is

stated, similar to this work. Lio et al. (2021) show wake position tracking with simultaneous estimation of the deficit shape and shear profile. It is based on a few-beam staring lidar and an EKF considering the wake meandering dynamics. A RMSE of 0.05 $D$, 0.12 $D$, 0.18 $D$ is shown, for 5 %, 10 %, 15 % TI respectively. In their work, the reference is a 1Hz least-squares fit of a parametrized deficit to 178-point scans by three synchronized lidar WindScanners. The tracking is slightly more precise than in this work, while Lio's method is based on different inputs and requires less external information.

## 4.3 Applicability

Load-based and lidar-based wake estimation techniques have different outlooks for application. While lidars are still in the early industrial adaption phase, load-based approaches can be a reliable alternative and implemented using solely the standard sensors of modern wind turbines. This comes at the cost of slightly reduced observability, or dependency on external information. The accuracy of the load-based tracking also needs to be ranged in relation to the expected magnitude of wake deflections due to

wake steering control. At very short turbine spacing, such as in this experiment, the uncertainty of the EKF estimate is close to the expected magnitude of wake deflections Jiménez et al. (2009). The conclusion is, that purely using the wake position as closed-loop feedback is a too narrow consideration. Still, this paper shows that satisfying wake estimation with the ability to support robust closed-loop wake steering with suitable feedback information of high spatial and temporal resolution is possible. The time resolution helps especially when not only considering the absolute wake position estimate (which might

e.g. be corrupted by an aberated wake deficit), but the change in wake position, which can be the intended response to a wake-steering maneuver. The required knowledge of the ambient conditions can arguably be estimated by a front row wind turbine (Soltani et al., 2013). Wake steering is, contrary to active wake control using a de-rating strategy, mostly applicable in low turbulent stable situations. This is also where the methodology shows best performance in the field, in agreement to expectations according to simulations.

This paper focuses on the wake position as an exemplary aspect of wind condition awareness. A similar data set and processing chain could also be used to validate wind farm models or wind field reconstruction based on in-situ probings. Future work will be to embed sensor based estimation within dynamic wind farm models (Becker et al., 2022; Lejeune et al., 2022), thus to couple analytic models, e.g. for the wake deficit or wake deflection. The relevance of open-loop approaches in wind farm flow control persists, since the impact of every control action is delayed by the advection duration of wakes. Closed-loop

yaw control does not necessarily need to happen at very fast time scales, where its effect could overlap with those of wake meandering. State estimation could also support as a feedback, whether the open-loop models predict as expected, or (online) re-calibration is necessary (see Hulsman et al. (2024)). Furthermore, the estimation of wake constellations yields information



of the farm-effective wind direction. It can complement error-prone and point-probing nacelle vane signals, thus contribute to a consensus-based farm-effective wind direction (Annoni et al., 2019).

## 5 Conclusions

The paper presents a quantitative field comparison of two independently applied wake centre estimation methods: a nacelle-based scanning lidar and an EKF based on rotor loads. The methodology accounts for the fact that there is no ground truth in the field by a detailed uncertainty evaluation. The lidar estimates have an uncertainty in the order of $0.05\,D$, which is a suitably precise reference to draw conclusions regarding the load-based EKF. It is a step forward in spatial resolution in comparison to the assumption of wake propagation parallel to the main wind direction. Both tracking methods agree with a RMSE of $0.2\,D$ for low to moderate TI, while increasing to $0.3\,D$ for a TI above $12\,\%$. The EKF formulation further yields the uncertainty of the state estimate as a byproduct, thus it self-indicates how certain or 'observable' a situation is. Insights to the full flow field with the lidar allow to identify the observability limits of the load-based EKF, e.g. to distinguish between the influence of exact wake shape and the wake centre position. Due to close turbine spacing and frequent high shear conditions, the observed wake deficits in this work are rather complex, mainly double-Gaussian, often asymmetric, and thus also influencing the wake centre definition. Yet, the methodology shows consistent behaviour even under these circumstances, which gives rise to the expectation that it would work similarly well or even better in settings with single-Gaussian wakes, although the wake load foot prints are relative weaker under such circumstances. While this paper focuses on the wake position as one aspect of wind condition awareness, it also outlines how wind farm models or turbine-based wind field reconstruction can be validated with complementary lidar data.





## Appendix A: Determination of rotor angle via edgewise blade root loads

The rotor angle of WT2 was not available to the authors. It was reconstructed from the edgewise blade root bending moments $M_e$. The method is based on the assumption that gravity is the main force contributing to the variation of the edgewise blade root bending moment (RBM). Let $\Psi$ be the rotor angle, which is defined positive clockwise and zero for blade 1 pointing upwards. Accordingly, each blade's edgewise RBM is modeled as

$$M_{e,1} = \hat{M}_e \sin(\Psi) + \bar{M}_e \qquad M_{e,2} = \hat{M}_e \sin(\Psi + \frac{2\pi}{3}) + \bar{M}_e \qquad M_{e,3} = \hat{M}_e \sin(\Psi + \frac{4\pi}{3}) + \bar{M}_e \,, \tag{A1}$$

where $\hat{M}_e$ is the amplitude and $\bar{M}_e$ is a non-oscillating offset, connected to the rotor torque. Using the addition theorem of the sine function

$$\sin(a+b) = \sin(a)\cos(b) + \sin(b)\cos(a) \,, \tag{A2}$$

this can be reformulated as

$$\begin{bmatrix} M_{e,1} \\ M_{e,2} \\ M_{e,3} \end{bmatrix} = \begin{bmatrix} 1 & 0 & 1 \\ -\frac{1}{2} & \frac{\sqrt{3}}{2} & 1 \\ -\frac{1}{2} & -\frac{\sqrt{3}}{2} & 1 \end{bmatrix} \begin{bmatrix} \hat{M}_e \sin(\Psi) \\ \hat{M}_e \cos(\Psi) \\ \bar{M}_e \end{bmatrix} \quad \Leftrightarrow \quad \begin{bmatrix} \hat{M}_e \sin(\Psi) \\ \hat{M}_e \cos(\Psi) \\ \bar{M}_e \end{bmatrix} = \begin{bmatrix} 1 & 0 & 1 \\ -\frac{1}{2} & \frac{\sqrt{3}}{2} & 1 \\ -\frac{1}{2} & -\frac{\sqrt{3}}{2} & 1 \end{bmatrix}^{-1} \begin{bmatrix} M_{e,1} \\ M_{e,2} \\ M_{e,3} \end{bmatrix} \,. \tag{A3}$$

The rotor angle is calculated as

$$\Psi = \mathrm{atan}\left( \frac{\hat{M}_e \sin(\Psi)}{\hat{M}_e \cos(\Psi)} \right) \,. \tag{A4}$$

To mitigate impact of the blade loads not behaving purely harmonic, e.g. due to tower shadow and non-uniform inflow, the nominator and denominator in Equation (A4) are filtered with a zero-phase low pass filter. The filtering at this point avoids filtering a non-continuous angle signal. The resulting determination of the rotor angle is shown for field data in Figure A1 and shows the expected behavior. The method was additionally validated with the aeroelastic model of the turbine in openFAST, where the rotor angle is available.

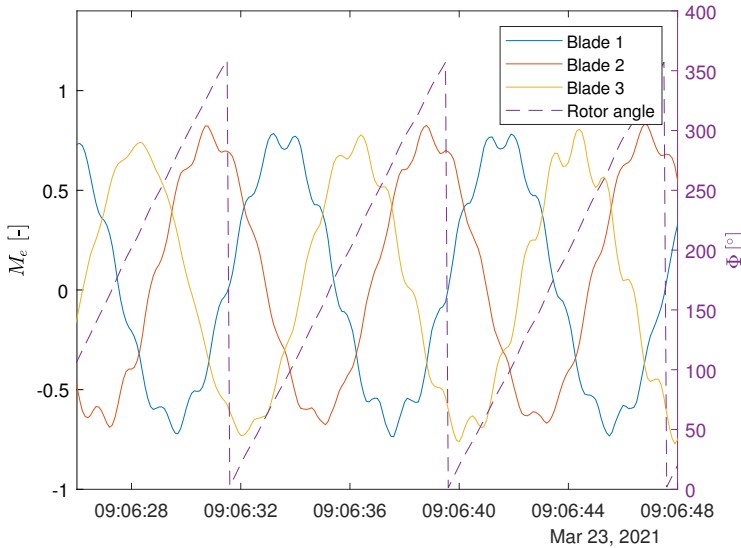

**Figure A1.** Identification of the rotor angle (on right y-axis) from edgewise blade loads (on left y-axis); loads were non-dimensionalized for confidential reasons

*Author contributions.* DO conceptualized, processed the data, generated the results, and wrote and edited the manuscript. GCL, AWHL, MK and VP provided intensive consultation on the conceptual design and generation of the results. PH organized and executed the lidar and wake steering campaign. VP had a supervisory function. All co-authors reviewed the manuscript.

*Competing interests.* The authors declare that there is no competing interest.

*Acknowledgements.* The authors would like to thank Eno Energy Systems GmbH, specifically Alexander Gerds, for the opportunity to carry out these experiments on their commercial turbines. The field campaign was conducted within the national research project *CompactWind II* (ref. no. FKZ 0325492H). The research was further supported by the national research project *DFwind - Phase 1* (ref. no. FKZ 0325936C). Both projects were funded by the Federal Ministry for Economic Affairs and Climate Action (BMWK) on the basis of a decision by the German Bundestag. Stephan Stone, Anantha Padmanabhan Kidambi Sekar and Jörge Schneemann are thanked for helping during the installation of the sensors. Raghawendra Manoj Joshi supported the creation of the FASTfarm model. Balthazar Arnoldus Maria Sengers, Andreas Rott and Lucy Pao are thanked for fruitful discussions.





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
