# Peer review of "Field comparison of load-based wind turbine wake tracking with a scanning lidar reference"

_Wind Energy Science, 2024_

## Referee Comment (RC1)

**Review of the publication**
**Field comparison of load-based wind turbine wake tracking with a scanning lidar reference**

Authored by David Onnen, Gunner Christian Larsen, Alan Wai Hou Lio, Paul Hulsman, Martin Kühn, and Vlaho Petrović

Dear authors,

I enjoyed reviewing your well-written and relevant publication on the wake location estimator. One aspect I really value is the consideration of uncertainty and the resulting bounds. My criticism is mainly related to the methodology presentation. Some sections would benefit from some clarification and additional information.

With kind regards

**Comments**

**Introduction**

The introduction is, in my opinion, a bit too brief on related work and not formulated consistently. Please review the phrasing and connection of the different sentences to generate a better reading flow. In contrast, I think the discussion (section 4) is well written, especially the paragraph starting at Line 438, which connects this work well with similar ones.

I am missing the motivation to use an extended Kalman Filter. Why not apply an Ensemble Kalman Filter or an Unscented Kalman Filter? The paragraph starting with line 36 lists several works that have done some sort of tracking, but I am missing a phrase more insight into how these publications have solved the issue, how the publications are connected, and what their successes and shortcomings are. I am also missing the link to the, in my opinion, relevant publications
*Towards the multi-scale Kalman filtering of dynamic wake models: observing turbulent fluctuations and wake meandering*, R. Braunbehrens et al 2023
and
*Closed-loop coupling of a dynamic wake model with a wind inflow estimator*, J. Di Cave et al 2024

Line 24 - „encountered" seems an odd choice. Do you mean „accounted for"?
Line 30 - Missing citation

**Methodology**

Line 64 - Missing citation

Line 78 - Wind shear exponent at 50Hz is used for what?

Line 98 - How does the height difference between the turbines affect this setup?

Line 109 - Is the assumption of zero mean justified? How are the matrices Q and R populated? Was some sort of normalization necessary?

Line 141-143 Please add a source or rephrase to make clear where this statement comes from.

Line 150 - The dynamic system for the wake center is, in principle, a random walk model. And, while a random walk's value is zero, an individual random walk is also expected to travel further away from the origin. Translated to the wake center, I would expect the model to be somewhat stable within a given region - if the wind direction does not change, we would expect the wake to meander within given bounds, e.g., +- 2D. This is even more the case for the z component, where we expect the wake to be within a narrower corridor.
Can the equations easily be adapted to incorporate this behavior? One approach could be to adapt Eq. 8a) ( and 8b), respectively) to
$\dot{y}_w(t) = v_c(t) - k\, y_w(t) + n_{x,1}(t),$
where k is a feedback constant. However, the change would cause the meandering around the origin, which can then be offset with a changing reference.
To be clear, I think the chosen approach is valid if the system is continuously corrected. I just wonder if you do see the same limitations of the model, or if I am missing something?

Line 179 - What are typical values for „b,c,d"? Do they have a major contribution or are they minor compared to the rest?

Line 181 - Can you elaborate on $M_{max}$ and $R_{mix}$? What do they represent, and how do you determine them?

Section 2.2.3 would strongly benefit from a figure to illustrate the different moments and angles, possibly also in connection with the incoming wake and the thereby resulting moments. The text is a bit tricky to follow the way it is written right now.

Section 2.2.3 should further emphasize the link between the states introduced in Section 2.2.2 and the output. Line 161 briefly mentions equation h(x,n) but then doesn't mention it again.

Equation 9-12 are a bit confusing to me: (9) introduces a method to calculate $M\_yaw$, $M\_tilt$ and $M\_col$ based on sensor data, (10) then discusses how to get $M^\wedge\sim(r\_w)$, just to invert it to return a different way of also calculating $M\_yaw$ and $M\_tilt$, followed by (12) which then tells the reader how to calculate $M^\wedge\sim(r\_w)$.
I think what you are missing is that the $M\_yaw$ and $M\_tilt$ from Eq(10) and (11) are estimates based on the estimate of $M^\wedge\sim(r\_w)$, which is based on the estimated states. If this is the case, please adapt the notation with the (^) symbol and think about reversing the derivation:
States -> r_w and \theta -> $M^\wedge\sim(r\_w)$ -> M_yaw and M_tilt
Maybe also add a similar block diagram to Fig. 2 with a more detailed flow of the signals.

Line 214 - Review the grammar of the sentence

Line 220 - I'd expect a list of the fitted parameters here / insight into the derived LuT.

Line 229 - I assume this is the azimuth of the lidar? Since the rotor azimuth was already defined with a variable in the previous sections. Maybe add a word to clarify that. Addition: Table 2 confirms that it's the Lidar azimuth; just change it here.

Equation 15 / 16 - Are \gamma_1 and \gamma_2 already defined?

Section 2.3.1 would profit from a sketch showing the different coordinate systems in relation to each other. This also makes it easier to interpret the results later on.

Line 243 - Is this an issue in the comparison to the load-based approach? Both aim to determine the wake center but at different positions. This question is also related to the fact that the turbines seem to have different heights (as indicated in Section 2.1).

Line 244 - Missing citation

Figure 5 has a very brief caption; I'd add where the data is coming from (lidar, I assume). Also, indicate the wind direction.

Section 2.3.3 / Table 2 How are the uncertainties defined? Are the \pm values upper and lower bounds or standard deviations?

**Results**

Figure 9 - Based on the explanation of „Geometry" I would expect it to be a line / some sin or cos. However, around (205 deg, 150 m), the scattering shows a spread, the same for the other end of the data. How come?

Line 311 - There is no figure supporting the claim of the asymmetry during yawed conditions. Consider adding a second figure to Figure 11 with the data.

Line 313 - I suggest to remove the „However".

Figure 12 - The jet/rainbow colormap leads to severe misrepresentation of data and should not be used. For more information, see Figure 3 h) in *The misuse of colour in science communication*, Crameri et al. 2020, https://www.nature.com/articles/s41467-020-19160-7
Note that WES also cites this publication in their submission information https://www.wind-energy-science.net/submission.html#figurestables

Figure 12 - a)-e) These are some of the main results of your paper, I'd increase the size of the figures significantly and add the then current wake location estimate. Consider removing double y-Axis for instance to get more space

Line 335 - Is it worth to add a subplot to Figure 12 with the yaw angle of WT1, and the wind direction? Additionally, Figure 12 does not indicate where the wake would be if it wasn't deflected due to the wake steering. If you add the geometric reference, this will become more visible.

**Discussion**

Section 4.2 contains a lot of comparisons. Is it possible to visually put them next to each other? It might make it easier to see if there is a common trend or significant differences.

I am missing a Data & Code availability statement.

---

## Editor Comment (EC1)

**Review of the paper "Field comparison of load-based wind turbine wake tracking with a scanning lidar reference" by Onnen et al.**

The paper deals with an investigation into the accuracy of a Kalman filter-based wake center tracking strategy using real field data. The topic is already addressed in literature but now the Authors were able to compare the tracking outputs with a reference, which is expected to be more accurate, i.e. a scanning Lidar.

The topic is worth investigating and the work well executed. The manuscript is clear and well-organized. I recommend publishing it. At the same time, I suggested some minor corrections and comments. Among all, I consider those listed under the section "Important comments" as something that, if addressed, may significantly improve the manuscript.

Important comments:

1. Page 6, equations (8): much is written in relation to the cut-out frequency of the low-pass filter that models the wake meandering, but very little is said about the adequacy of the model. Can Authors enlarge the treatment, possibly including a reference? Moreover, is it possible to detail how the Authors considered the variability of mean wind speed $u_\infty$ and turbulence intensity in such a model?

2. Page 9, equation (9): Please, notice that transforming blade loads through the Coleman transformation yields two pieces of information (M_tilt and M_yaw) really close to the nodding and yawing moments that are easier to measure (e.g. strain-gauges on main bearing). Surely, they are not identical (e.g. the nodding moment there will be biased due to rotor weight) but they should carry the very same informative content requested by the detector. Given the fact that "The rotor azimuth angle information of WT2 was not available" (see line 74), this consideration could be practically relevant rather than a pure mathematical comment. Please comment.

3. Line 170: "The yaw and tilt moment depend on the wake position"; this is true, but they depend on other parameters, such as the shear layer magnitude. Authors cope with this by adding the terms $b$ and $c$ in eq. (11), to model, among all, also the impact of shear. However, the shear is variable too. Can Authors comment on this fact?

4. Line 210 and subsequent: important considerations. Good to see them here. Can the Authors provide insight into the possible application of the methodology using field data where one cannot control and decide a priori the inflow conditions to use to train the model?

Minor comments:

1. Line 30 and 64: missing references.
2. Figure 9: consider increasing plot dimensions.

3. Figure 9b: consider the possibility to add a new figure, representing the error between "Geometry" and "Geometry + Jimenez" versus the lidar estimate. This could improve the interpretation of the results.
4. Section 3.1.2: at what downstream distance is the speed deficit measured by the Lidar?
5. Comments on Fig.12: it is important to notice that the estimator is able to detect wake impingement on both sides of the rotor (left/right). I totally understand that maybe Authors considered it self-evident or trivial, but this is the very first capability that a wake detector must have.

---

## Author Comment (AC1)

**Response to reviewer's comments - Reviewer 1**

Authors:         David Onnen, Gunner Chr. Larsen, Wai Hou Lio, Paul Hulsman, Martin Kühn, Vlaho Petrovic

Paper Number:  WES-2024-188

Title:            Field comparison of load-based wind turbine wake tracking with a scanning lidar reference

Color coding:    Reviewer comments, authors responses, paper citations

Dear authors,
I enjoyed reviewing your well-written and relevant publication on the wake location estimator. One aspect I really value is the consideration of uncertainty and the resulting bounds. My criticism is mainly related to the methodology presentation. Some sections would benefit from some clarification and additional information.

Dear Reviewer,
First and foremost, thank you for taking the time to read through and review our manuscript. Answering your comments increased the quality of the manuscript. Thank you also for your positive words about the consideration of uncertainties and the writing of the manuscript. In the following we address each of your comments individually.

With kind regards,
The authors

**Introduction**

1) The introduction is, in my opinion, a bit too brief on related work and not formulated consistently. Please review the phrasing and connection of the different sentences to generate a better reading flow. In contrast, I think the discussion (section 4) is well written, especially the paragraph starting at Line 438, which connects this work well with similar ones.

The introduction section was revised, especially in regard to the phrasing. Further information on related work was added. The discussion section you highlighted is still the place, where the detailed comparison to other methodologies' results takes place. That allows the reader to compare and range the results of this work and the related works under consideration of the respective methodologies. In order to avoid double-mentioning, a reference to the discussion section was added in the introduction. The changes to the introduction section can be seen in the diff-document.

2) I am missing the motivation to use an extended Kalman Filter. Why not apply an Ensemble Kalman Filter or an Unscented Kalman Filter? The paragraph starting with line 36 lists several works that have done some sort of tracking, but I am missing a phrase more insight into how these publications have solved the issue, how the publications are connected, and what their successes and shortcomings are. I am also missing the link to the, in my opinion, relevant publications

*Towards the multi-scale Kalman filtering of dynamic wake models: observing turbulent fluctuations and wake meandering*, R. Braunbehrens et al 2023
and
*Closed-loop coupling of a dynamic wake model with a wind inflow estimator*, J. Di Cave et al 2024

The Extended Kalman Filter (EKF) was sufficient for this application - it is the simple-most and computationally most efficient form here. Note, that the dynamic model is already formulated as a linear system, thus only the measurement model needs a local linearization (this aspect is now added to section 2.2.1 explicitly). Reasons to use an Ensemble Kalman Filter (EnKF) or an Unscented Kalman Filter (UKF) would be a higher-dimensional problem or strong non-linearity of the model. Other authors works, e.g. (Becker et al., 2022) and (Braunbehrens et al., 2023) use wind farm models with a high number of observation points (>100). In these cases, the EnKF becomes more efficient than the local linearization with respect to each of the states.

Note, that the state transition model $f$ used in this work can be formulated as a linear operation (see next subsection). Thus, the local linearisation in Equation 4 is not necessary in every iteration - $\mathbf{F_k}$ can be directly pre-computed.

The reference to (Braunbehrens et al., 2023) was added, both here and in 2.2.2 (regarding the distinction between time scales, together now with the work of (Rott et al., 2018; Simley et al., 2020)). The information contained in (J. Di Cave et al 2024) was considered covered by the works of Becker et al.

3) Line 24 - „encountered" seems an odd choice. Do you mean „accounted for"?

Indeed, your suggestion sounds more suitable. It was implemented.

While robust formulations can account for wind direction variability (Rott et al., 2018; Simley et al., 2020), optimal wake deflection cannot be guaranteed, since outer influences and wake dynamics can hardly be accounted for.

4) Line 30 - Missing citation

Please excuse the inconvenience of this and thank you for pointing it out. It turned out to be a corrupted bibtex item that slipped our checks for the final compilation of the document. We have of course corrected this in the revised version for all occurrences of this reference (thus we will not address this aspect for the following instances individually in this authors response). The missing reference was (Kidambi Sekar et al., 2024).

**Methodology**

5) Line 78 - Wind shear exponent at 50Hz is used for what?

We apologize for the confusion. Of course, a wind shear exponent at 50 Hz is neither necessary nor senseful. "All so far mentioned measurements are stored at 50Hz" was said at this point, because the description of the lidar measurements with different sampling frequencies begins in the following. We changed the order of sentences to avoid any confusion.

[…] Both the turbine and met the mast data is stored at 50Hz.

The wind shear exponent $\alpha$ is calculated from the met mast measurements according to the power law: [...]

6) Line 98 - How does the height difference between the turbines affect this setup?

The height difference is accounted for in the generation of training data of the load-based estimator. The FASTfarm model uses the correct heights of the individual turbines. Assuming purely horizontal propagation of the wake, the lidar would probe the wake location at a slightly lower altitude than the hub height of WT2. Yet, the convolution method is not expected to be notably influenced regarding the lateral wake position it returns. Since only single PPI lidar scans are available, the exact vertical wake position cannot be addressed here. It is, however, also not in focus of wind farm flow control.

7) Line 109 - Is the assumption of zero mean justified? How are the matrices Q and R populated? Was some sort of normalization necessary?

Kalman filters by definition handle zero-mean white noise, thus the statement in Line 109 is to be seen as the plain definition of variables to be used within the filter. Since the wake dynamics are not modelled with white noise, the state augmentation is done as described in section 2.2.2, realizing the necessary noise shaping while maintaining the original Kalman filter equations. **Q** and **R** are diagonal matrices. Normalization needs to be considered for the dynamic model if the methodology is applied at a different sampling frequency. The state transition covariance of course rises, if a larger time increment is on hand. Further details on the noise tuning can be checked in (David Onnen et al., 2023) in an idealized environment and for a non-commercial turbine (thus not subject to confidentiality aspects regarding the loads).

8) Line 141-143 Please add a source or rephrase to make clear where this statement comes from.

This also refers to the DMW model by (Larsen et al., 2008), as described in the preceding two sentences. It is said now explicitly.

Wake meandering in the atmospheric boundary layer is driven by turbulence patterns considerably larger than the wake deficit scale (Trujillo et al., 2011). Larsen et al. (2008) introduced the DWM model, which translates this split of scales to a random walk trajectory, where the wake deficit is seen as a passive tracer. Larsen et al. define the default cut-off frequency of the meandering motion is defined as $f_c = u_\infty/(2D_w)$, where $D_w$ is the wake diameter (in near wake applications also the rotor diameter $D$ is a valid choice). Note, that this is the theoretical limit, up to which a wake deficit is regarded as a passive tracer. Lio et al. (2021) show in a field study with a lidar-based EKF featuring an auto-correlation term of the wake position time history that the dominant spectral share of the meandering motions can be up to a factor 10 slower.

9) Line 150 - The dynamic system for the wake center is, in principle, a random walk model. And, while a random walk's value is zero, an individual random walk is also expected to travel further away from the origin. Translated to the wake center, I would expect the model to be somewhat stable within a given region - if the wind direction does not change, we would expect the wake to meander within given bounds, e.g., +- 2D. This is even more the case for the z component, where we expect the wake to be within a narrower corridor.
Can the equations easily be adapted to incorporate this behavior? One approach could be to adapt Eq. 8a) ( and 8b), respectively) to   \dot{y}_w(t) = v_c(t) - k y_w(t) + n_{x,1}(t),

where k is a feedback constant. However, the change would cause the meandering around the origin, which can then be offset with a changing reference.
To be clear, I think the chosen approach is valid if the system is continuously corrected. I just wonder if you do see the same limitations of the model, or if I am missing something?

Thank you for sharing your thoughts and impulse on this topic! We agree with your statement "The dynamic system for the wake center is, in principle, a random walk model. And, while the expected value of the random walk is zero, an individual random walk can be expected to diverge from the origin." The model formulation, however, is not considering an individual random walk. Instead, it describes the probability distribution of the wake dynamics via the additive process noise covariance. And a "self-correction" is achieved by the Kalman filter as you suggest, by including measurements at every iteration. Additionally, since the wind direction (and turbine yaw) can change to a constellation of ceasing wake impingement, it is in fact possible that the wake moves laterally "out of bounds" of WT2, such that it is not observable anymore.
The formulation with a reverting term that you suggest would formulate an Ornstein-Uhlenbeck (OU) process. This would be a valid choice indeed, for situations where the wake position meanders around zero or around a mean position. This would be the case in simulation environments or wind tunnel applications with a constrained wind direction. An OU process is in fact used in (D Onnen et al., 2024) to synthesize wake trajectories for artificial wake conditions in a wind tunnel. Note however, that the spectra of the OU process and a random walk is congruent, such that no implications on the observer formulation in this paper are on hand.
Including a mean-reverting term for the vertical wake position could be worth considering in future, to further improve the robustness of the estimation. The onset should however be checked in a test environment, where a vertical wake position reference is available.

10) Line 179 - What are typical values for „b,c,d"? Do they have a major contribution or are they minor compared to the rest?

These are additional tuning parameters for the consistent description of the model equations. We cannot state the absolute values for the load offsets $b$ and $c$ here, but we can say that the onset is small in comparison to the wake-induced aerodynamic load imbalances. The parameter $d$, describing the yaw-tilt-coupling has a typical value of a few degrees (<10°).

11) Line 181 - Can you elaborate on M_max and R_mix? What do they represent, and how do you determine them?

M_max is the maximum amplitude of the yaw/tilt moment with respect to the wake position. This maximum is given when the wake is at distance $r_w$ = R_mix, or in the 1D case $y_w$ = R_mix. They are determined by fitting the parameter model to the training data. This explanation was added to the manuscript with the overview on the fitting parameters in Table 1 (see response to question 16).

12) Section 2.2.3 would strongly benefit from a figure to illustrate the different moments and angles, possibly also in connection with the incoming wake and the thereby resulting moments. The text is a bit tricky to follow the way it is written right now.

Thank you for this feedback. We now indicate the fitting parameters now in Figure 3 and Figure 4, which illustrate the parametric model (see response to question 16).

13) Section 2.2.3 should further emphasize the link between the states introduced in Section 2.2.2 and the output. Line 161 briefly mentions equation h(x,n) but then doesn't mention it again.

We have restructured the section to make this link more conclusive now.

The measurement model $h$ is a mapping from the state to the measurement - in this study a link from the wake centre position to the rotor loads. The model must fulfill certain criteria: It should be computationally inexpensive, such that it can be computed online in each filter iteration. Look-up tables with pre-computed information are preferable here, see e.g. (Schreiber et al., 2020; Soltani et al., 2013)). Moreover, the model has to be differentiable, such that its local sensitivity to a change in state or input can be determined. Finally, it should be robust and lead to a convergence of the estimate, even if the state at initialization is far off. The measurement vector $y_k$ contains the Coleman transformed, non-rotating flapwise blade root bending moments according to Eq. 9. The time index $k$ is omitted from the notation for better readability.

[…]

In the following, the parameterised model is derived in Equations 10-13.  All fitting parameters introduced in this scope are listed in Table 1. The model is subsequently fitted to training data generated in aeroelastic simulations with enabled DWM model. Figure 3 shows the contour shape of the model and Figure 4 an example of training data and fitting.

[…]

14) Equation 9-12 are a bit confusing to me: (9) introduces a method to calculate M_yaw, M_tilt and M_col based on sensor data, (10) then discusses how to get M^~(r_w), just to invert it to return a different way of also calculating M_yaw and M_tilt, followed by (12) which then tells the reader how to calculate M^~(r_w). I think what you are missing is that the M_yaw and M_tilt from Eq(10) and (11) are estimates based on the estimate of M^~(r_w), which is based on the estimated states. If this is the case, please adapt the notation with the (^) symbol and think about reversing the derivation: States -> r_w and \theta -> M^~(r_w) -> M_yaw and M_tilt
Maybe also add a similar block diagram to Fig. 2 with a more detailed flow of the signals.

By definition, the Kalman filter compares measurements y with the measurement estimates y_hat = h(x_hat,u). It is, however, not common practise to use the (^) symbol within the formulation of the measurement model h() (see e.g. (Brown & Hwang, 1992; Lio et al., 2021; Soltani et al., 2013)). We would like to keep the order of the derivation. But we added another clarification after Eq. 9, stating that the modelling starts here (see previous comment).

15) Line 214 - Review the grammar of the sentence

The sentence was adjusted.

For other cases, e.g. for larger streamwise spacing, a higher-dimensional LUT is required to adequately resolve the impact of turbulent mixing in the far-wake region.

16) Line 220 - I'd expect a list of the fitted parameters here / insight into the derived LuT.

Thanks for pointing this out. It was added.

**Table 1.** Fitting parameter for the measurement model $h()$

| Parameter | Unit | Description |
|---|---|---|
| $R_{\mathrm{mix}}$ | m | wake overlap resulting in the largest yaw/tilt moment; an approximation is $R_{\mathrm{mix}} = (R + R_{\mathrm{w}})/2$, i.e. the mean of rotor radius and wake deficit radius |
| $\tilde{M}_{\mathrm{max}}$ | Nm | maximal value of yaw/tilt moment (reached at wake overlap $R_{\mathrm{mix}}$) |
| $b$ | Nm | wake-independent offset of yaw moment |
| $c$ | Nm | wake-independent offset of tilt moment |
| $d$ | ° | phase angle to describe yaw-tilt-coupling |
| $M_0$ | Nm | collective moment at full wake overlap |
| $M_\infty$ | Nm | collective moment at no wake overlap |

17) Line 229 - I assume this is the azimuth of the lidar? Since the rotor azimuth was already defined with a variable in the previous sections. Maybe add a word to clarify that. Addition: Table 2 confirms that it's the Lidar azimuth; just change it here.

Yes, this is the lidar azimuth angle. It was adjusted as you suggest.

18) Equation 15 / 16 - Are \gamma_1 and \gamma_2 already defined?

Thanks for pointing this out. We added a definition prior to Table 2.

The nacelle yaw angles are denoted $\gamma_1$ and $\gamma_2$ for WT1 and WT2, respectively.

19) Section 2.3.1 would profit from a sketch showing the different coordinate systems in relation to each other. This also makes it easier to interpret the results later on.

We have added a sketch of the coordinate systems as you suggest.

[Figure]

**Figure 5.** Illustration of the coordinate systems as defined in Table 2

20) Line 243 - Is this an issue in the comparison to the load-based approach? Both aim to determine the wake center but at different positions. This question is also related to the fact that the turbines seem to have different heights (as indicated in Section 2.1)

The impact on the comparison is assumed minimal. Assuming advection velocities of 5-10 m/s, the time difference between the upstream probing and the onset on the rotor is at the order of 10-20 seconds. Meanwhile, a lidar sample is recorded within 30s and the cut-off frequency of the wake dynamics for the EKF is 0.01 Hz. Thus, the impact of the different probing locations vanishes in comparison to the involved time scales of the wakes and their estimation.
It is true that we cannot compare at the exactly same positions, as it would be that case e.g. in an aeroelastic simulation with a DWM wind field that can be checked isolated for the wake position. We chose the best compromise by probing as close to the turbine as possible while also not being affected by the induction zone. This is the inaccessible reference in line 244, which investigates the induction zone in the same wind farm: (Kidambi Sekar et al., 2024). It describes how stream tube widening around the turbine leads to a lateral flow component, which redirects incoming partial wakes outwards.

Regarding the turbine height, please refer to our answer to comment 6.

21) Figure 5 has a very brief caption; I'd add where the data is coming from (lidar, I assume). Also, indicate the wind direction.

Thank you for pointing this out. The information on the data source (lidar indeed) and wind direction was added.

Figure 5. Wake centre identification from lidar measurements in WT2-based coordinate system. The wind direction here is 228° resulting in a full wake constellation.

22) Section 2.3.3 / Table 2 How are the uncertainties defined? Are the \pm values upper and lower bounds or standard deviations?

This is stated in the caption of Table 2 (now Table 3): "[…] values relate to the 95% confidence interval for normally distributed uncertainties". This means, that a coverage factor of 2 is used, so the $\pm 2\sigma$ bounds.

23) Figure 9 - Based on the explanation of „Geometry" I would expect it to be a line / some sin or cos. However, around (205 deg, 150 m), the scattering shows a spread, the same for the other end of the data. How come?

That is really well spotted. The general expectation of the sine behaviour for the "Geometry" scatters is very well fulfilled. The small scattering you point out relates to instances of steep wind direction change, which is followed by a delayed yaw action of WT2. Since the wake position $y_w$ is expressed in WT2-based coordinates, the deviation occurs. We consider no impact for the results of this paper.

24) Line 311 - There is no figure supporting the claim of the asymmetry during yawed conditions. Consider adding a second figure to Figure 11 with the data.

Thank you for bringing this up. Due to your comment, we have had another look at the binning with respect to yaw misalignment. It turned out that when applying high thresholds on the yaw misalignment (>10°), a small trend is visible indeed: While the main asymmetry of the double Gaussian deficit, i.e. the magnitude difference of the two wake peaks, is still mainly linked to the ambient shear, we see a tendency towards a broader peak at the pronounced side of the wake at negative yaw misalignment. This finding is to be treated with care, since it is based on small data availability (compare Figure 9). We thus don't see enough evidence to make a generalized claim here, but the behaviour is in line with what we would expect according to literature on wakes of misaligned turbines (Bartl et al., 2018; Bromm et al., 2018; Sengers et al., 2020). The deviations we see could also explain for the slight RMSE increase of the tracking in case of negative yaw misalignments (see Figure 16), since the wake deficits slightly stand out from the others. We have done the following changes to include this to the paper:
- adding the plot to Figure 11 (now Figure 12) as you suggest
- describing the finding in section 3.1.2
- including the aspect to the discussion in section 4.1

[Figure]

**Figure 12.** Wake deficits within wind speed bin $7.5 - 8\,\mathrm{m\,s^{-1}}$; a) colour coded for two ranges of shear profile, defined by power law coefficient $\alpha$; b) colour coded with respect to yaw misalignment of WT1

In section 3.1.2

The co-occurence of the asymmetry with ambient conditions is documented in Figure 12. A strong impact is visible when filtering for the power law coefficient $\alpha$, describing the shear profile. Figure 12a indicates that the wake asymmetry is more pronounced at strong shear, connected to atmospheric stable conditions. For low shear coefficients, the wake deficits are rather symmetric. Larger wind speed variations among the deficits as well as in the non-waked area are on hand here, which again is attributed to the atmospheric stability. Figure 12b shows a distinction of wake deficits with respect to yaw-misalignment situations, which are known to cause a kidney-shaped curled wake (see e.g. Bartl et al., 2018; Sengers et al., 2023). While the main asymmetry of the double Gaussian deficit, i.e. the magnitude difference of the two wake peaks, is linked to the ambient shear, a tendency towards a broader peak at the pronounced side of the wake is seen in case of negative yaw misalignment. This finding is to be treated with care, since it is based on small data availability (compare Figure 9). The role of the wake deficit in this context is further discussed in section 4.1.

In section 4.1

The wake asymmetry is found to dominantly co-occur with strong wind shear and to increase with ambient wind speed, and thus also rotational speed. An interaction of wake rotation and the sheared flow is assumed. The rotational component in the wake flow, in opposite direction to the rotor rotation, could cause an `upwash' of wind speeds from low altitudes on the right side of the rotor (facing downstream, thus negative on the y-axis) and a `downwash' of wind speeds from higher altitudes on the left side. The direction of wake rotation and the observed orientation of the wake asymmetry would support this explanation. A comparable near wake asymmetry is reported by (Bromm et al., 2018) in a similar field campaign. A minor co-occurence of wake asymmetry and large WT1 yaw misalignments (>10°) is found, matching the expectation with regard to the curled wake phenomena (Bartl et al., 2018; Sengers et al., 2023). Yet, data availability of large yaw misalignments is not considered sufficient to draw a clear conclusion on curled wakes, which are also not in focus of this work.

25) Line 313 - I suggest to remove the „However"

The section was reformulated following the previous comment.

26) Figure 12 - The jet/rainbow colormap leads to severe misrepresentation of data and should not be used. For more information, see Figure 3 h) in *The misuse of colour in science communication*, Crameri et al. 2020, https://www.nature.com/articles/s41467-020-19160-7
Note that WES also cites this publication in their submission information https://www.windenergy-science.net/submission.html#figurestables

Thanks for addressing the topic and hinting at the reference. We have changed the colormaps of Figures 5,6,12 (now Fig. 6,7,13) accordingly. The recommended colormap 'viridis' is used.

[Figure]

**Figure 6.** Wake centre identification from lidar measurements in WT2-based coordinate system. The wind direction here is 205°, resulting in a partial wake constellation.

[Figure]

**Figure 7.** Illustration of uncertainty propagation: Probe position uncertainty in $x, y, z$ and horizontal wind speed uncertainty; WT2-based coordinate system

[Figure]

**Figure 13.** Top: Time series of wake position estimate by load-based EKF and lidar; the uncertainty range for both methods is indicated
Bottom: Snapshots of the instantaneous flow situation in the wind farm; ground-based coordinates are used; WT1 indicated in black, WT2 in red; the time instances a-e refer to the indications in the time series plot on top

27) Figure 12 - a)-e) These are some of the main results of your paper, I'd increase the size of the figures significantly and add the then current wake location estimate. Consider removing double y-Axis for instance to get more space.

Thank you for pointing this out. The y-axis was removed for all but the leftmost snapshot, such that the figure size is increased. An example indication of the wake position estimate from lidar is given in Figure 6, which uses the WT2-based coordinate system. The visualisation in Figure 12 (now Figure 13) uses the ground-based coordinates for better overview of the general flow situation and turbine constellation.

28) Line 335 - Is it worth to add a subplot to Figure 12 with the yaw angle of WT1, and the wind direction? Additionally, Figure 12 does not indicate where the wake would be if it wasn't deflected due to the wake steering. If you add the geometric reference, this will become more visible.

Regarding the first aspect: The ambient conditions are shown in Figure 14, including the wind direction. The yaw misalignment of WT1 was added to the plot.
Regarding the second aspect: The geometric reference was intentionally not added in the top plot of Figure 12 (now Figure 13) for two reasons: i) as discussed in the context of Figure 10, the pure consideration of the farm geometry and the assumption of wake propagation parallel to the main wind direction is not capturing the wake position variability accurately. Also, occurrences of wake steering are just one aspect causing the wake position variability. ii) Adding another signal (plus its respective uncertainty range) would make the plot quite messy and distract from the main

comparison of this paper as mentioned in the title. Three overlapping uncertainty intervals with shading could hardly be told from one another.

29) Section 4.2 contains a lot of comparisons. Is it possible to visually put them next to each other? It might make it easier to see if there is a common trend or significant differences.

As elaborated, we consider a direct comparison of all approaches in literature not meaningful, since very different test scenarios and performance metrics are used. This is exactly the reason for this detailed section, addressing all approaches individually, under specific consideration of their respective setting and metric.

30) I am missing a Data & Code availability statement.

Thanks for pointing this out! The statement was added.

Bartl, J., Mühle, F., Schottler, J., Sætran, L., Peinke, J., Adaramola, M., & Hölling, M. (2018). Wind tunnel experiments on wind turbine wakes in yaw: Effects of inflow turbulence and shear. *Wind Energy Science*, *3*(1), 329–343. https://doi.org/10.5194/wes-3-329-2018

Becker, M., Allaerts, D., & van Wingerden, J. W. (2022). Ensemble-Based Flow Field Estimation Using the Dynamic Wind Farm Model FLORIDyn. *Energies*, *15*(22), 1–23. https://doi.org/10.3390/en15228589

Braunbehrens, R., Tamaro, S., & Bottasso, C. L. (2023). Towards the multi-scale Kalman filtering of dynamic wake models: observing turbulent fluctuations and wake meandering. *Journal of Physics: Conference Series*, *2505*(1), 012044. https://doi.org/10.1088/1742-6596/2505/1/012044

Bromm, M., Rott, A., Beck, H., Vollmer, L., Steinfeld, G., & Kühn, M. (2018). Field investigation on the influence of yaw misalignment on the propagation of wind turbine wakes. *Wind Energy*, *21*(11), 1011–1028. https://doi.org/10.1002/we.2210

Brown, R. G., & Hwang, P. Y. C. (1992). Introduction to random signals and applied kalman filtering (second edition), Robert Grover Brown and Patrick Y. C. Hwang, John Wiley, New York, 1992, 512 p.p., ISBN 0–47152–573–1, $62.95. In *International Journal of Robust and Nonlinear Control* (Vol. 2, Issue 3). https://doi.org/10.1002/rnc.4590020307

Kidambi Sekar, A. P., Hulsman, P., Van Dooren, M. F., & Kühn, M. (2024). Synchronised WindScanner field measurements of the induction zone between two closely spaced wind turbines. *Wind Energy Science*, *9*(7), 1483–1505. https://doi.org/10.5194/wes-9-1483-2024

Larsen, G. C., Madsen, H. A., Thomsen, K., & Larsen, T. J. (2008). Wake meandering: A pragmatic approach. *Wind Energy*, *11*(4), 377–395. https://doi.org/10.1002/we.267

Lio, W. H., Larsen, G. C., & Thorsen, G. R. (2021). Dynamic wake tracking using a cost-effective LiDAR and Kalman filtering: Design, simulation and full-scale validation. *Renewable Energy*, *172*, 1073–1086. https://doi.org/10.1016/j.renene.2021.03.081

Onnen, D, Neuhaus, L., Petrović, V., Ribnitzky, D., & Kühn, M. (2024). Dynamic wake conditions

tailored by an active grid in the wind tunnel. *Journal of Physics: Conference Series*, *2767*(4), 042038. https://doi.org/10.1088/1742-6596/2767/4/042038

Onnen, David, Petrović, V., Neuhaus, L., Langidis, A., & Kühn, M. (2023). Wind tunnel testing of wake tracking methods using a model turbine and tailored inflow patterns resembling a meandering wake. *2023 American Control Conference (ACC)*, 837–842. https://doi.org/10.23919/ACC55779.2023.10155916

Rott, A., Doekemeijer, B., Seifert, J. K., van Wingerden, J.-W., & Kühn, M. (2018). Robust active wake control in consideration of wind direction variability and uncertainty. *Wind Energy Science*, *3*(2), 869–882. https://doi.org/10.5194/wes-3-869-2018

Schreiber, J., Bottasso, C. L., & Bertelè, M. (2020). Field testing of a local wind inflow estimator and wake detector. *Wind Energy Science*, *5*(3), 867–884. https://doi.org/10.5194/wes-5-867-2020

Sengers, B. A. M., Steinfeld, G., Heinemann, D., & Kühn, M. (2020). A new method to characterize the curled wake shape under yaw misalignment. *Journal of Physics: Conference Series*, *1618*(6). https://doi.org/10.1088/1742-6596/1618/6/062050

Sengers, B. A. M., Steinfeld, G., Hulsman, P., & Kühn, M. (2023). Validation of an interpretable data-driven wake model using lidar measurements from a field wake steering experiment. *Wind Energy Science*, *8*(5), 747–770. https://doi.org/10.5194/wes-8-747-2023

Simley, E., Fleming, P., & King, J. (2020). Design and analysis of a wake steering controller with wind direction variability. *Wind Energy Science*, *5*(2), 451–468. https://doi.org/10.5194/wes-5-451-2020

Soltani, M. N., Knudsen, T., Svenstrup, M., Wisniewski, R., Brath, P., Ortega, R., & Johnson, K. (2013). Estimation of rotor effective wind speed: A comparison. *IEEE Transactions on Control Systems Technology*, *21*(4), 1155–1167. https://doi.org/10.1109/TCST.2013.2260751

Trujillo, J.-J., Bingöl, F., Larsen, G. C., Mann, J., & Kühn, M. (2011). Light detection and ranging measurements of wake dynamics. Part II: two-dimensional scanning. *Wind Energy*, *14*(1), 61–75. https://doi.org/10.1002/we.402

---

## Author Comment (AC2)

**Response to reviewer's comments - Reviewer 2**

Authors: David Onnen, Gunner Chr. Larsen, Wai Hou Lio, Paul Hulsman, Martin Kühn, Vlaho Petrovic

Paper Number: WES-2024-188

Title: Field comparison of load-based wind turbine wake tracking with a scanning lidar reference

Color coding: Reviewer comments, authors responses, paper citations

Dear Reviewer,
First and foremost, thank you for taking the time to read through and review our manuscript. Answering your comments increased the quality of the manuscript. In the following we address each of your comments individually.

With kind regards,
The authors

1) While kind of discussed, it would help to state your contributions to this work explicitly in a pointwise list at the end of the introduction.

If you refer to the authors contributions – these are documented at the end of the paper, after the appendix and before the references. The scientific contribution to the research field is indeed documented at the end of the introduction, just before the paper structure is outlined. Here we define the research gap and deduce how this work fills that gap. We have adjusted the part to highlight the sub-aspects of the research gap that are answered in this paper.

"The research gap can be concluded as follows: Existing work for load-based wake tracking lacks either

- a consideration of wake dynamics and time resolution, or
- a field validation, or
- (in case of a field validation) an independent reference to compare with.

The objective of this work is to fill the gap by addressing all three aspects: The works shows direct estimation of the instantaneous wake centre position in a field experiment with two utility-scale wind turbines. The load-based estimate is compared to the wake position probed with a scanning lidar, which serves as an independent reference. To that purpose, the uncertainty of the lidar estimate is quantified using analytic error propagation following the GUM […]."

2) Sect 2.1: You directly start describing the wind farm, while I would expect it would be more interesting to say something first about the scientific contribution you are bringing with your work. Consider changing the order.

We have indeed considered a different order, even when writing the initial draft, but chose to stay with this order for two reasons. Firstly, we see the field testing itself as one of the core scientific

contributions of this paper, so no contradiction in starting with description of the field experiment. Secondly, the field setup influences the wake estimation methodologies. A general idea of the setup is important to understand certain follow-up topics, e.g. how the training data for the load-based method is generated, how the lidar-related coordinate transforms are formulated or which sensor uncertainties need to be considered.

3) Fig XX: All figures need a more elaborate caption. Now the figures are not interpretable apart from the main text.

Thank you for pointing this out. We have added additional information to the captions of most figures to make them more self-explanatory.

4) Eq 3 to 7: very standard theory, really needed to include in this paper? Or make it more specific to your case. Also, explain why you assume 0 noise acting on the state and output.

We agree that the formulation of an EKF is well-known for people from a control & estimation background. For people from a wind physics or lidar background it might be new, that is why we introduce it to make sure we document our work steps thoroughly. We see the same for other papers in this field, e.g. (Braunbehrens et al., 2023; Eichstadt et al., 2016; Lio et al., 2021).

Regarding your second point: This must be a misunderstanding, we do not assume zero noise acting on the state and output. The notation of the models $f$ and $h$ implies to define the noise as an input, as seen e.g. in Eq.8. When using the local linearisations in the EKF, however, the noise term enters via the additive noise covariance matrices $\mathbf{Q}$ and $\mathbf{R}$ (see Eq. 4&5). Thus, the second input to the models $f$ and $h$ is set to zero (to not conflict with the notation while also not implying another source of noise).

The equations are discretized for their implementation in the state transition function $f(x_k, n_{x,k})$. Note that the $n_{x,i}$ represents the $i^{th}$ element of the noise vector $n_x$. The time index $k$ is omitted here, because the continuous representation is chosen. Since the noise term enters linearly, they are incorporated in the EKF formulation via the additive noise covariance matrix $\mathbf{Q}$.

5) Eqs 8a-8d: Please elaborate more on this model. It seems very simple for the dynamics you want to capture. Is it linear? If yes, why do you need an EKF, and not a normal KF? Also, elaborate more about how a (linear?) combination of the chosen state vector elements leads to the 3 nonrotating blade moments. An elaborate explanation and justification of the dynamic model and chosen measurements are largely missing. ---> Ah, you explain this in the next subsection. Would it make sense to swap the order 2.2.3 and 2.2.2? So first fully define f() and h(), and then incorporate them into the state estimator.

Regarding the state transition model described in Eqs. 8a-8d:
The entire section 2.2.2 motivates and derives the state transition model, leading to the formulation in Eq. 8. As you have noticed, the loads are not touched here, because the state transition model solely considers how the filter states change with time. This is mentioned at the beginning of section 2.2.2:
The dynamic model describes how the system state evolves over time. In this study, the model should capture how the wake centre position changes over time. Depending on the atmospheric conditions and the wind farm control strategy, the wake trajectory is subject to various dynamic

influences. Time scales of wind direction changes, wake-steering control and wake meandering need to be incorporated by the dynamic model of the EKF, while effects corresponding to small-scale turbulence with no expressiveness towards the wake position need to be rejected.

Regarding the linearity of the model:
As you point out correctly, the dynamic model $f(x_k, n_{x,k})$ to describe the random walk behaviour of a meandering wake indeed boils down to a first-order linear formulation. However, the measurement transition model $h(x_k, n_{y,k})$ (described in section 2.2.3) is nonlinear. That is why an EKF is required. Still, $f(x_k, n_{x,k})$ does not need to be linearized in every iteration. Instead, the state transition matrix **F** can be formulated directly. We added this information explicitly to the general EKF formulation in section 2.2.1

Note, that the state transition model $f(x_k, n_{x,k})$ used in this work can be formulated as a linear operation (see next subsection). Thus, the local linearisation in Eq.4 is not necessary in every iteration, since **F** can be directly pre-computed.

Regarding the order:
Section 2.2.1 describes the EKF formulation and defines the state and measurement vectors. This needs to happen first, otherwise the models f() and h() would not be interpretable.

Section 2.2.2 defines the state transition function f(). It appears first in the EKF algorithm and is also more concise, that is why we describe it first.

Section 2.2.3 defines the measurement transition function h(). It links the wake position to the rotor loads. We see no benefit in swapping order with section 2.2.2.

Section 2.2 describes the aforementioned structure, such that the reader knows in which order the load-based wake tracking is presented. We added more detail to the description of the section sturcuture at the beginning of sect. 2.2 in order to avoid any confusion for the reader.

The interaction between the individual aspects of the load-based wake tracking problem is shown in the overview chart in Fig.2. The EKF and its sub-components are described in the following sections. In section 2.2.1 the EKF formulation and the definition of states and inputs takes place. Section 2.2.2 defines the state transition function *f()*, and section 2.2.3 defines the measurement transition function *h()*.

6) 2.2: Kind of a literature survey. Can it be largely moved to the introduction of the paper?

Section 2.2 is not a literature survey. We assume that you refer to section 2.2.2, which includes a number of literature references. In section 2.2.2, the state transition model is motivated and we provide background information for the formulation of Eq.8. As you argued in your previous comment, such an elaboration is desirable to explain the design choices. We discuss specific aspects of wake and wind direction dynamics and their characteristic time scales. This has a direct relation to the model formulation of this section. In our opinion, it is far too detailed and extended to find room in the general introduction section of this paper.

7) 2.2.3: You use the Coleman transformation to obtain the nonrotating blade moments (tilt/yaw). It is well-known that for larger, more flexible rotors, you need some sort of decoupling strategy -- possibly in the Coleman transformation by an azimuth offset -- to obtain decoupled axes. You do seem to consider this aspect with the variable "d". Because it is a crucial aspect for larger flexible

rotors, I highly recommend that you incorporate it into your research and elaborate more; there have been publications on this topic in the past.

You are perfectly right, the yaw-tilt coupling needs to be considered and this is in fact done via the parameter *d*. It describes the phase delay of an inert blade reaction when fed through the Coleman transform. In the context of this work we do not state absolute values of *d* since it refers to a commercial turbine subject to confidentiality. We further added the references (Lu et al., 2015) and (Mulders et al., 2019) which provides background information on the yaw-tilt-coupling in high detail. Moreover, the parameter *d* is now indicated in the contour plot of the measurement model:

[Figure]

**Figure 3.** Contour plot of the measurement model outputs in dependency of the wake position. Normalized with their respective maximum and minimum for confidentiality. The fitting parameter $d$ is indicated, describing the phase-offset of the yaw-tilt-coupling.

8) Sect 3.: I got lost in the structure of this section. Please announce what you will be discussing in the first part of the section (directly under 3.), and come up with a clearer structure, so that the storyline makes more sense.

Thank you for pointing this out. Please note that the general paper structure is outlined at the end of the introduction. But we understand the need to give guidance at the beginning of each main section. We thus added the following description in the beginning of section 3.

In this section, the results of the field experiment and the wake estimation are reported. In section 3.1, the wake conditions contained in the data set are described, considering both the wake position variability and the wake deficit shape. In section 3.2, the wake position estimates of the load-based EKF and the lidar are compared.

9) Sect 4.: Also, what is the purpose of this section? What will you discuss? Announce that at the start of the section.

Also thanks for hinting at this. We added the following description in the beginning of section 4.

In this section, the results are interpreted and ranged. First, the influence of the site specifications on the results is discussed, considering the generalizability of the findings. Secondly, the wake tracking performance is discussed. The comparison to existing works in literature considers their individual testing conditions and performance metrics. Finally, the applicability of the presented wake tracking in the context of wind condition awareness and wind farm flow control is discussed.

10) Often, a "?" appears when citing, check

Please excuse the inconvenience of this and thank you for pointing it out. It turned out to be a corrupted bibtex item that slipped our checks for the final compilation of the document. We have of course corrected this in the revised version for all occurrences of this reference. The missing reference was (Kidambi Sekar et al., 2024).

Braunbehrens, R., Tamaro, S., & Bottasso, C. L. (2023). Towards the multi-scale Kalman filtering of dynamic wake models: observing turbulent fluctuations and wake meandering. *Journal of Physics: Conference Series*, *2505*(1), 012044. https://doi.org/10.1088/1742-6596/2505/1/012044

Eichstadt, S., Makarava, N., & Elster, C. (2016). On the evaluation of uncertainties for state estimation with the Kalman filter. *Measurement Science and Technology*, *27*(12), 125009. https://doi.org/10.1088/0957-0233/27/12/125009

Kidambi Sekar, A. P., Hulsman, P., Van Dooren, M. F., & Kühn, M. (2024). Synchronised WindScanner field measurements of the induction zone between two closely spaced wind turbines. *Wind Energy Science*, *9*(7), 1483–1505. https://doi.org/10.5194/wes-9-1483-2024

Lio, W. H., Li, A., & Meng, F. (2021). Real-time rotor effective wind speed estimation using Gaussian process regression and Kalman filtering. *Renewable Energy*, *169*, 670–686. https://doi.org/10.1016/j.renene.2021.01.040

Lu, Q., Bowyer, R., & Jones, B. L. (2015). Analysis and design of Coleman transform-based individual pitch controllers for wind-turbine load reduction. *Wind Energy*, *18*(8), 1451–1468. https://doi.org/10.1002/we.1769

Mulders, S. P., Pamososuryo, A. K., Disario, G. E., & Wingerden, J. van. (2019). Analysis and optimal individual pitch control decoupling by inclusion of an azimuth offset in the multiblade coordinate transformation. *Wind Energy*, *22*(3), 341–359. https://doi.org/10.1002/we.2289

---

## Referee Report (RR1)

**Review of wes-2024-188**

**Field comparison of load-based wind turbine wake tracking with a scanning lidar reference**

Authored by: David Onnen, Gunner Chr. Larsen, Alan W. H. Lio, Paul Hulsman, Martin Kuhn and Vlaho Petrovic

Dear Authors,

I very much enjoyed reading and reviewing your paper. I think it is a very good piece of work, worthy of publication in the *Wind Energy Science* Journal after minor revisions. Especially, I have found the results section very nice. The different events captured by the Lidar and EKF are well explained. In contrast, the Methodology section could deserve more clarifications. I would thus suggest revising mostly the clarity and accuracy in the methodology section, as per the major and minor comments below. If these comments can be addressed, I support a publication in WES.

Best regards,

Reviewer

**Major comments:**

Sect. 1 Introduction: the literature review could deserve to be expanded. I think the list of prior works is good and complete, but some more sentences for each justifying the difference with the present work could be good, to understand better the novelty brought here already from this introduction part.

Sect. 2.2.1 General EKF setup: you start already defining many abstracts variables and mathematical model, before the main problem has been even clarified, as:

- The physics involved (wake deficit, dynamic wake meandering, wake deflection, etc.)
- The quantities that must be estimated (wake position etc.) and why they are relevant for which applications.
- The main inputs that you use (blade root loads yaw tilt col) and theoretical explanation why they include the relevant information you try to predict.

  I think a subsection clarifying these points would be very helpful before the current Sect. 2.2.1.

Line 212-213: *"Only one stochastic seed per wind field proved sufficient, since the effect of ambient turbulence is low in comparison to the effect of the wake."* I think this is a very dangerous and misleading statement, since ambient turbulence directly affects the effect of the wake when one considers dynamic environment. It is a core element of the DWM model that the ambient turbulence is the main driver of the whole wake propagation and dynamic meandering. Hence, various turbulent seeds can produce very different wake effects based on the DWM. I am quite skeptical that a single seed is enough for convergence. Please justify this statement more in detail, ideally with adding numerical tests and convergence study (possibly as Appendix). Furthermore, when comparing synthetic simulation data with field measurements (as the topic of this paper), extra attention must be given to seed-to-seed variability and binning approaches for measurements data, to obtain statistically converged data. Please elaborate on this.

In Sect. 2.2.1 you present the state vector as four parameters (yw, zw, vc and wc). Yet, in the whole results part you only show predictions of the lateral wake position (yw). I miss the part where you justify why you only look at yw for the results. Especially because this wind farm has the particularity of having two different hub heights, results on the vertical wake position (zw) could be very interesting to include (and for the application perspective, the vertical position is as important as the lateral one).

**Minor comments:**

Sect. 2.1 Field experiment: I think there should be proper citations added for each of the measurement devices mentioned (Trimble type 3 Zephyr mode, Thies Clima type 4.3352.00.400, Leosphere WindCube 200S, etc.) in the references.

Sect. 2.1 last paragraph: you mention the active wake steering control applied on WT1 but it would be great to have the yaw schedule added as a plot here for more clarity (scheduled yaw angle of WT1 by wind direction).

Sect. 2.2 a proper citation for EKF is missing.

Line 284: probably a typo (Myaw, Myaw) twice.

Line 244. Typo reference (?)

Fig. 15: This figure is nice, but it is a bit misleading to represent both metrics as parallel bars, since the RMSE should be as low as possible and the inRange should be as high as possible. Please consider possible review of this point (possibly by redefining the metric "inRange" into "NotInRange" so that it should also be as low as possible). The figure would thus be much easier to interpret in my opinion.

Thanks!

---

## Referee Report (RR2)

- Section 2.2.1 General EKF setup: It is not inherently clear how the Extended Kalman Filter bridges the gap between the measurements and the calculated wake position. Please either provide a reference or explain the Extended Kalman Filter in more detail. In particular, equations (3) – (7) are not explained at all and the variables $f$ and $h$ are introduced in the text but not explained. How are the gradients for $F_k$ and $H_k$ computed? Maybe use another variable than $f$ here, as this is used for the frequency throughout the rest of the paper.
- Section 2.2.2 Dynamic model: The sentence in line 147 'The meandering time scales are thus incorporated with a first-order expression' is confusing. Please explain what exactly you mean by this and show an equation.
- Section 2.3.1 Coordinate System: The offset in equation (16) corresponds to the 2.7D the turbines are spaced from each other. This is however only mentioned in section 2.1. Please add a sentence here to explain where this offset is coming from.
- Section 2.3.3 Equation (19) is mathematically incorrect, as a vector cannot be squared. Is this meant in a computational sense, i.e. each component of the vector is supposed to be squared? Please specify.
- Section 3.2 Please explain the variable $\Omega$ in Equation (22)
- There are question marks instead of citations in some places (Lines: 30, 64, 244)

---

## Author Response (AR2)

**Response to reviewer's comments - Reviewer 3**

Authors:          David Onnen, Gunner Chr. Larsen, Wai Hou Lio, Paul Hulsman, Martin Kühn,
                  Vlaho Petrović

Paper Number:  WES-2024-188

Title:            Field comparison of load-based wind turbine wake tracking with a scanning lidar
                  reference

Color coding:     Reviewer comments, authors responses, paper citations

Dear Authors,
I very much enjoyed reading and reviewing your paper. I think it is a very good piece of work,
worthy of publication in the *Wind Energy Science* Journal after minor revisions. Especially, I
have found the results section very nice. The different events captured by the Lidar and EKF
are well explained. In contrast, the Methodology section could deserve more clarifications. I
would thus suggest revising mostly the clarity and accuracy in the methodology section, as
per the major and minor comments below. If these comments can be addressed, I support a
publication in WES.
Best regards,
Reviewer

Dear Reviewer,
First and foremost, thank you for taking the time to read through and review our manuscript.
Answering your comments increased the quality of the manuscript. In the following we address each
of your comments individually.

With kind regards,
The authors

**Major Comments**

1) Sect. 1 Introduction: the literature review could deserve to be expanded. I think the list of prior
works is good and complete, but some more sentences for each justifying the difference with
the present work could be good, to understand better the novelty brought here already from
this introduction part.

Thank you for pointing out this aspect. The introduction section was revised. The key novelties to
close the research gap are highlighted in bullet points. Furthermore, a reference to the discussion
section was added in the introduction, where the outcome of our work is compared and ranged with
regard to existing work.

[…] The research gap can be concluded as follows: Existing work for load-based wake tracking lacks
either

- a consideration of wake dynamics and time resolution, or
- a field validation, or
- (in case of a field validation) an independent reference to compare with.

The objective of this work is to fill the gap by addressing all three aspects: The works shows direct estimation of the instantaneous wake centre position in a field experiment with two utility-scale wind turbines. The load-based estimate is compared to the wake position probed with a scanning lidar, which serves as an independent reference. To that purpose, the uncertainty of the lidar estimate is quantified using analytic error propagation following the GUM.

[…]

In section 4, the findings are discussed, ranged and compared with literature.

2) Sect. 2.2.1 General EKF setup: you start already defining many abstracts variables and mathematical model, before the main problem has been even clarified, as:
• The physics involved (wake deficit, dynamic wake meandering, wake deflection, etc.)
• The quantities that must be estimated (wake position etc.) and why they are relevant for which applications.
• The main inputs that you use (blade root loads yaw tilt col) and theoretical explanation why they include the relevant information you try to predict.

I think a subsection clarifying these points would be very helpful before the current Sect. 2.2.1.

The general problem statement takes place at the end of the introduction section.

The work shows direct estimation of the instantaneous wake centre position in a field experiment with two utility-scale wind turbines.

In section 2.2, the estimation is initially outlined, before moving to the EKF formulation in 2.2.1. We initially provide the general estimation scheme for overview, thus also the definition of states and measurements. The involved physics are described along with their respective modeling in 2.2.2 and 2.2.3. That way, repetitions can be avoided, even though it requires introducing variables early, without discussing them in detail yet.

We extended in 2.2 for guidance:

Core of the tracking algorithm is an Extended Kalman Filter (EKF), which links the load measurements from a wake-exposed wind turbine with the physical knowledge about the wake dynamics. […]
In section 2.2.1, the EKF formulation and the definition of states and inputs takes place. Section 2.2.2 defines the state transition function f(), including a consideration of the involved wake physics. Section 2.2.3 defines the measurement transition function h(), so the linkage between wake position and rotor loads.

Addressing the 2nd and 3rd point, information was added in the introduction section.

In order to increase the spatial observability of non-uniform turbine inflow, the rotor imbalances - resulting from shear, yaw misalignment or wake impingement - can be encountered (Bertelè et al., 2017). These rotor imbalances, such as yaw- and tilt-moments, are related to the harmonics of the blade root bending moments. The Coleman transform describes the translation from the rotating to the non-rotating coordinate system.
Ultimately relevant for wake-steering control is the wake position within the wind farm, which is the feature that a wind farm controller aims to manipulate. Existing methods for the wake position estimation are […]

The applicability is further discussed in section 4.3.

3) Line 212-213: *"Only one stochastic seed per wind field proved sufficient, since the effect of ambient turbulence is low in comparison to the effect of the wake."* I think this is a very dangerous and misleading statement, since ambient turbulence directly affects the effect of the wake when one considers dynamic environment. It is a core element of the DWM model that the ambient turbulence is the main driver of the whole wake propagation and dynamic meandering. Hence, various turbulent seeds can produce very different wake effects based on the DWM. I am quite skeptical that a single seed is enough for convergence. Please justify this statement more in detail, ideally with adding numerical tests and convergence study (possibly as Appendix). Furthermore, when comparing synthetic simulation data with field measurements (as the topic of this paper), extra attention must be given to seed-to-seed variability and binning approaches for measurements data, to obtain statistically converged data. Please elaborate on this.

We see your point and how the highlighted sentence is misleading. In short: we fully agree with your statement regarding the ambient turbulence as the driver of wake meandering and the according implementation in the DWM. We also agree that a different seed will produce a different wake trajectory for a given 10min simulation. For the generation of training data, however, the exact trajectory is not of interest. The meandering helps to populate the spreading of wake positions in the training data in addition to the subsequent lateral shift of the wake-causing turbine WT1 (see Fig. 4). Accordingly, the model fitting at one ambient condition already combines the results of seven simulations with their own wind field (referring to the seven lateral WT1-positions, indicated with different colors in Fig.4).
Our statement regarding the effect of ambient turbulence also addresses the effect of background turbulence.

We have adjusted the section as follows:

Thus, it is decided to only create training data in dependency of the ambient wind speed, resulting in a 1-dimensional lookup-table (LUT) of fitting parameters. This requires 63 simulations (7 WT1 positions and 9 wind speeds, 4-12m/s), each with a duration of 600s, a TI of 10 % and α=0.25. Only one stochastic seed per wind field proved sufficient, since the set for one ambient condition already combines the results of seven simulations with their respective wind field (referring to the seven lateral WT1-positions).

4) In Sect. 2.2.1 you present the state vector as four parameters (yw, zw, vc and wc). Yet, in the whole results part you only show predictions of the lateral wake position (yw). I miss the part where you justify why you only look at yw for the results. Especially because this wind farm has the particularity of having two different hub heights, results on the vertical wake position (zw) could be very interesting to include (and for the application perspective, the vertical position is as important as the lateral one).

In fact, we do consider the vertical position less relevant for the application, because
a)    it has lower position variance due to meandering (see e.g. (Braunbehrens & Segalini, 2019)) and is less affected by wind direction changes
b)    it cannot be manipulated by wake-steering control

Furthermore, no reference for the vertical wake position is available from the lidar measurements, thus the vertical position could not be compared.

The aspect is justified in section 2.2:

[…] Note, that the estimation task is here formulated for the general, 2-dimensional case, so considering the horizontal and vertical wake position. Due to the measurement setup and the single PPI scans of the lidar, only a comparison of the horizontal component is possible, which is also more relevant. The vertical position is considered less relevant for the application, because *i)* it has lower position variance due to wind direction changes and meandering (Braunbehrens & Segalini, 2019), and *ii)* it cannot be manipulated by wake-steering control.

**Minor comments**

5) Sect. 2.1 Field experiment: I think there should be proper citations added for each of the measurement devices mentioned (Trimble type 3 Zephyr mode, Thies Clima type 4.3352.00.400, Leosphere WindCube 200S, etc.) in the references.

It was added as you suggest.

Thies-Clima: Data Sheet - Wind Direction Transmitter 4.3151.xx.40x, https://www.thiesclima.com/en/db/dnl/4.3151.xx.40x_wr-geber-firstclass_eng.pdf, 2025a.

Thies-Clima: Data Sheet - Wind Transmitter 4.3352.00.4xx, https://www.thiesclima.com/pdf/en/first-class--ice-classified/wind-transmitter-first-class-advanced-x$\sim$4.3352.00.4xx/, 2025b.

Trimble: Data Sheet - Zephyr 3 GNSS Antenna, https://geonovus.ee/wp-content/uploads/pdf/Datasheet-TrimbleZephyr3.pdf, 2025.

Trujillo, J.-J.: Large scale dynamics of wind turbine wakes, Dissertation, 2017.

Trujillo, J.-J., Bingöl, F., Larsen, G. C., Mann, J., and Kühn, M.: Light detection and ranging measurements of wake dynamics. Part II: two-dimensional scanning, Wind Energy, 14, 61–75, https://doi.org/10.1002/we.402, 2011.

Vaisala: Data Sheet - Windcube 100S/200S/400S, https://www.vaisala.com/sites/default/files/documents/Windcube100_200_400s_3D-Doppler-Lidar-Brochure_WC_BD.pdf, 2025.

Vollmer, L., Steinfeld, G., Heinemann, D., and Kühn, M.: Estimating the wake deflection donstream of a wind turbine in different atmospheric stabilities: An LES study, Wind Energy Science, 1, 129–141, https://doi.org/10.5194/wes-1-129-2016, 2016.

6) Sect. 2.1 last paragraph: you mention the active wake steering control applied on WT1 but it would be great to have the yaw schedule added as a plot here for more clarity (scheduled yaw angle of WT1 by wind direction).

Our goal here is not to analyse wake steering, but to detect the wake position. The presence of wake steering controller makes the wake position more dynamic and challenging, but we feel that explaining the wake steering strategy running on WT1 would not benefit the paper.

7) Sect. 2.2 a proper citaton for EKF is missing.

We added this citation for the EKF in section 2.2: (Brown & Hwang, 1992)

8) Line 284: probably a typo (Myaw, Myaw) twice.

Thank you! As you noticed correctly, (Myaw, Mtilt) was meant. It is fixed now.

9) Line 244. Typo reference (?)

Thanks for spotting this. It is fixed in the latest version.

10) Fig. 15: This figure is nice, but it is a bit misleading to represent both metrics as parallel bars, since the RMSE should be as low as possible and the *inRange* should be as high as possible. Please consider possible review of this point (possibly by redefining the metric "*inRange*" into "*NotInRange*" so that it should also be as low as possible). The figure would thus be much easier to interpret in my opinion.

Thanks for the suggestion. However, we prefer keeping the definition as it is. The *inRange* metric allows for a better comparison to the detection ratio used in (Bottasso et al., 2018), as described in section 4.2.

In addition to the mathematical definition in Eq. 22, the figure caption reminds the reader:

[…] the orange bars refer to the right y-axis and represent the *inRange* indicator, so whether the difference between the position estimates is covered by their uncertainty intervals.

**References:**

Bottasso, C. L., Cacciola, S., & Schreiber, J. (2018). Local wind speed estimation, with application to wake impingement detection. *Renewable Energy*, *116*, 155–168. https://doi.org/10.1016/j.renene.2017.09.044

Braunbehrens, R., & Segalini, A. (2019). A statistical model for wake meandering behind wind turbines. *Journal of Wind Engineering and Industrial Aerodynamics*, *193*(August), 103954. https://doi.org/10.1016/j.jweia.2019.103954

Brown, R. G., & Hwang, P. Y. C. (1992). Introduction to random signals and applied kalman filtering. In *John Wiley* (Vol. 2, Issue 3). https://doi.org/10.1002/rnc.4590020307

**Response to reviewer's comments - Reviewer 4**

Authors:      David Onnen, Gunner Chr. Larsen, Wai Hou Lio, Paul Hulsman, Martin Kühn, Vlaho Petrović

Paper Number:  WES-2024-188

Title:           Field comparison of load-based wind turbine wake tracking with a scanning lidar reference

Color coding:     Reviewer comments, authors responses, paper citations

Dear Reviewer,
First and foremost, thank you for taking the time to read through and review our manuscript. Answering your comments increased the quality of the manuscript. In the following we address each of your comments individually.

With kind regards,
The authors

1) Section 2.2.1 General EKF setup: It is not inherently clear how the Extended Kalman Filter bridges the gap between the measurements and the calculated wake position. Please either provide a reference or explain the Extended Kalman Filter in more detail. In particular, equations (3) – (7) are not explained at all and the variables $f$ and $h$ are introduced in the text but not explained. How are the gradients for $Fk$ and $Hk$ computed? Maybe use another variable than $f$ here, as this is used for the frequency throughout the rest of the paper.

Thank you for pointing this out. We have done the following in regard to your points:

- We added the reference (Brown & Hwang, 1992) for the EKF in section 2.2.
- The role of each model parameter, used to link wake position and measurements, is described in Table 1.
- The notation f() and h() for the state transfer model and measurement transfer model is standard in literature – thus we would like to keep this. We added this part to introduce the notation:
  Section 2.2.2 defines the state transition function $f$(), and section 2.2.3 defines the measurement transition function $h$().
  Furthermore we ensure a distinction by denoting the cut-off frequency of the meandering $f_c$ and the frequency axis of the spectra in Figure 15 reads "frequency [Hz]" explicitly.
- Eq.3-7 form the standard procedure of the EKF algorithm, following Brown & Hwang. The linearization is explained in more detail:
  The local linearisations of the state transition model and the measurement model around a current state are denoted $\mathbf{F}_k$ and $\mathbf{H}_k$, respectively. They can generally be computed via forward Euler, following Equations 4,5. Note, that the state transition model f(x, n) used in this work can be formulated as a linear operation (see next subsection). Thus, the linearisation in Equation 4 is not necessary and $\mathbf{F}$ can be directly constructed from Equation 8.

2) Section 2.2.2 Dynamic model: The sentence in line 147 'The meandering time scales are thus incorporated with a first-order expression' is confusing. Please explain what exactly you mean by this and show an equation.

The term "first-order expression" refers to a first-order state-space model. The equation of motion for the wake dynamics considers dynamics up to the first derivative of the wake position, thus the crosswise velocity of the wake centre. The according equations are Eq.8a-d. We specified by changing the term to first-order differential equation:

The meandering time scales are thus modelled with first-order differential equations.

3) Section 2.3.1 Coordinate System: The offset in equation (16) corresponds to the 2.7D the turbines are spaced from each other. This is however only mentioned in section 2.1. Please add a sentence here to explain where this offset is coming from.

Thank you, it was added:

The x- and y-offsets in Eq. 16 refer to the 2.7D spacing in ground-based coordinates.

4) Section 2.3.3 Equation (19) is mathematically incorrect, as a vector cannot be squared. Is this meant in a computational sense, i.e. each component of the vector is supposed to be squared? Please specify.

Thank you for pointing this out! We have changed as follows:

Their propagation through the coordinate transform in Equations 14-16) is formulated by Equation 19, following the GUM standard (JCGM, 2020), where the expression $(.)^{\circ 2}$ denotes the element-wise square operation for a vector. Also, the square-root is to be understood element-wise. […]

$$\begin{bmatrix} \Delta x \\ \Delta y \\ \Delta z \end{bmatrix}_{WT2} = \sqrt{\left(\frac{\partial \boldsymbol{x}_{WT2}}{\partial \gamma_1}\Delta\gamma_1\right)^{\circ 2} + \left(\frac{\partial \boldsymbol{x}_{WT2}}{\partial \gamma_2}\Delta\gamma_2\right)^{\circ 2} + \left(\frac{\partial \boldsymbol{x}_{WT2}}{\partial \delta}\Delta\delta\right)^{\circ 2} + \left(\frac{\partial \boldsymbol{x}_{WT2}}{\partial \chi}\Delta\chi\right)^{\circ 2}}$$

5) Section 3.2 Please explain the variable Ω in Equation (22)

The variable Ω is solely used as a helper variable to ease the readability. It is defined within Equation 22 and matches the description in the paragraph:

[…] The additional metric inRange is introduced in Equation 22, denoting whether the estimates are within each other's 2σ uncertainty range.

6) There are question marks instead of citations in some places (Lines: 30, 64, 244)

This was fixed. We apologise for the inconvenience.

**Response to reviewer's comments - Reviewer 5**

Authors:        David Onnen, Gunner Chr. Larsen, Wai Hou Lio, Paul Hulsman, Martin Kühn, Vlaho Petrovic

Paper Number:  WES-2024-188

Title:            Field comparison of load-based wind turbine wake tracking with a scanning lidar reference

Color coding:    Reviewer comments, authors responses, paper citations

The paper deals with an investigation into the accuracy of a Kalman filter-based wake center tracking strategy using real field data. The topic is already addressed in literature but now the Authors were able to compare the tracking outputs with a reference, which is expected to be more accurate, i.e. a scanning Lidar.
The topic is worth investigating and the work well executed. The manuscript is clear and well organized. I recommend publishing it. At the same time, I suggested some minor corrections and comments. Among all, I consider those listed under the section "Important comments" as something that, if addressed, may significantly improve the manuscript.

Dear Reviewer,
First and foremost, thank you for taking the time to read through and review our manuscript. Answering your comments increased the quality of the manuscript. In the following we address each of your comments individually.

With kind regards,
The authors

**Important comments**

1) Page 6, equations (8): much is written in relation to the cut-out frequency of the lowpass filter that models the wake meandering, but very little is said about the adequacy of the model. Can Authors enlarge the treatment, possibly including a reference? Moreover, is it possible to detail how the Authors considered the variability of mean wind speed $u_\infty$ and turbulence intensity in such a model?

Thanks for this comment - that is an interesting aspect indeed. The topic relates to the split of scales regarding wind speed in the ABL, an aspect well described in (Soltani et al., 2013). In the scope of a rotor-effective wind speed estimator, authors decompose the wind speed into a mean component $\bar{u}$ and a turbulent component $u'$. The $u'$ is modelled as a Wiener process with coloured noise terms, $\bar{u}$ is considered the following way:

"The average wind speed must be able to vary slowly from zero to at least 30m/s. This is modelled by the simple random walk [...], where the incremental covariance [...] is set to $2^2/600$, i.e., the standard deviation of the average wind speed variation over 10 min is 2 m/s. " (Soltani et al., 2013)

The split of scales is defined around the scale ratio $\frac{\pi\,\bar{u}}{2L}$, where $L$ is the integral length scale. Considering commonly encountered values of $L$, the scale ratio has a similar order of magnitude as the cut-off frequency used in our work, which already justifies ignoring the turbulent part of more

complex dynamics, since it is below a scale with relevance for wake position shifts. The value of ambient mean wind speed is solely used to schedule the LUT parameter for the load model, described in section 2.2.3. No additional dynamics are considered here. The variability of rotor-effective wind speed of the wake-exposed turbine is probably best correlated with the collective load $M_{col}$. The measurement covariance entry of $M_{col}$ was fixed in this work. A future consideration, however, could be a self-tuning formulation, following the works of (Ritter, 2020).

Regarding the role of turbulence intensity: Ambient turbulence affects the meandering amplitude, the deficit shape and perturbations (see discussion section 4.2 'Impact of ambient conditions').

The observed increase of RMSE with TI is expected and agrees with simulation studies of load-based estimation (Dong et al., 2021; Onnen et al., 2022) and field results of lidar-based wake estimation (Lio et al., 2021). Higher turbulence intensities affect both the shape of the instantaneous wake deficit and the dynamics of the wake position. The information contained in the blade root loads is typically not sufficient to distinguish between both aspects, especially when their characteristic time scales are overlapping. The definition of the cut-off frequency in the dynamic model of the EKF leads to a rejection of turbulent scales smaller than the rotor scale. Deviations of the wake deficit shape that persist at scales of multiple rotor diameters could be misinterpreted as a change in wake position.

The impact of turbulence intensity on wake mixing can be included when creating higher-dimensional training data, which includes different TIs (see the answer to comment #4 below). Generally, simulations might allow better attempts for a parametric study. Please refer to (Onnen et al., 2022), where the role of turbulence intensity is checked an HAWC2 framework. The HAWC2 implementation of the DWM allows to distinguish ambient turbulence (responsible for wake mixing) and the large scale, meandering-driving turbulence field.

2) Page 9, equation (9): Please, notice that transforming blade loads through the Coleman transformation yields two pieces of information (M_tilt and M_yaw) really close to the nodding and yawing moments that are easier to measure (e.g. strain-gauges on main bearing). Surely, they are not identical (e.g. the nodding moment there will be biased due to rotor weight) but they should carry the very same informative content requested by the detector. Given the fact that "The rotor azimuth angle information of WT2 was not available" (see line 74), this consideration could be practically relevant rather than a pure mathematical comment. Please comment.

This is a very good idea and we have little doubt that such probing would redundantly contain the required information for a similar EKF. If bearing loads were generally available from strain gauges on the bearing or main shaft, or displacement sensors between bedplate and hub, it could be considered an alternative. A concern could be, that manufactures differ strongly regarding the drive train concepts thus different probing specifications needed to be accounted for and a generalizable approach might be difficult to achieve. Meanwhile, blade root bending moment probings leave less room for deviating concepts and are considered rather standard instrumentation on modern wind turbines. For retrofitting purposes on older turbines, however, easier instrumentation would be worth considering. A side note on the rotor azimuth angle: the value was not available in the data recordings, but normally it is available within the control environment of the turbine. When executing the estimator in real-time application on the turbine, it can be expected to be used.

3) Line 170: "The yaw and tilt moment depend on the wake position"; this is true, but they depend on other parameters, such as the shear layer magnitude. Authors cope with this by adding the terms $b$ and $c$ in eq. (11), to model, among all, also the impact of shear. However, the shear is variable too. Can Authors comment on this fact?

This is a fair point. We fully agree that the shear is variable and we are aware of the present simplification in the given example. The ambient wind shear mainly affects the tilt moment. Due to that, a different shear could be confused for a small shift in vertical wake position, when not fully accounted for by the fitting parameter $c$ (also $b$, considering yaw-tilt-coupling). A solution could be to create training data for multiple shears and create a higher-dimensional LUT that is scheduled with respect to ambient shear (this refers to your comment 4). The shear could be estimated by an upstream (non-waked) turbine. In the example we present, we are limited to assess the lateral wake position, because the single-elevation lidar scans do not grant a reference for the vertical position.

It was decided to not add further degrees of freedom to the training data in this paper, in order to not introduce further unknowns that might complicate the interpretation of the results. We suspect that the influence of shear on the wake deficit (e.g. wake asymmetry) is not fully represented in the DWM-implementation.

The respective paragraph in section 2.2.3 was extended.

The impact of shear on the wake deficit is not fully accounted for in the simulation environment, especially in relation to wake-asymmetry (as discussed later in section 4.1). Thus, it is decided to only create training data in dependency of the ambient wind speed, resulting in a 1-dimensional lookup-table (LUT) of fitting parameters. This requires 63 simulations (7 WT1 positions and 9 wind speeds, 4-12m/s), each with a duration of 600s, a TI of 10 % and $\alpha=0.25$. Only one stochastic seed per wind field proved sufficient, since the set for one ambient condition already combines the results of seven simulations with their respective wind field (referring to the seven lateral WT1-positions). Depending on the scenario, a higher-dimensional LUT can be required. A consideration of ambient TI is required in case of larger streamwise spacing, to adequately resolve the impact of turbulent mixing in the far-wake region. Also, including ambient shear could be a further step, preferably with a refined modeling of its impact on the wake deficit.

4) Line 210 and subsequent: important considerations. Good to see them here. Can the Authors provide insight into the possible application of the methodology using field data where one cannot control and decide a priori the inflow conditions to use to train the model?

Thanks for the comment. In principle, the spread of inflow conditions that cannot be known a priori can be covered by adding dimensions to the LUT. Thus, training data for a broader set of ambient conditions would be created and the model parameter scheduled with respect to these conditions (see previous comment), as it is already done with respect to ambient wind speed in the presented example. For an online-implementation, the ambient conditions then need to be estimated by another turbine, as shown e.g. in (Bertelè et al., 2021; Soltani et al., 2013). We suspect that one turbine alone struggles to simultaneously estimate both wakes and 'undisturbed' ambient conditions, due to limited observability. Coupling estimations from several turbines might be the best solution here, as discussed in section 4.3 'Applicability', referring e.g. to the works of (Becker et al., 2022).

**Minor comments**

1) Line 30 and 64: missing references.

Thanks for noticing and sorry for the error. It was fixed. The missing reference is: (Kidambi Sekar et al., 2024)

2) Figure 9: consider increasing plot dimensions.

Thanks for pointing this out. We will consider the dimensions and ensure readability when preparing for the double-column format used for final publication.

3) Figure 9b: consider the possibility to add a new figure, representing the error between "Geometry" and "Geometry + Jimenez" versus the lidar estimate. This could improve the interpretation of the results.

We see your point here. Yet, we deliberately decided to focus on the lidar reference in this paper. The take-away of Figure 9b is mainly: 'the lidar catches more wake position spread than suggested by the *steady* approaches "Geometry" and "Jimenez" '. The field assessment of the Jiminez model or wake trajectory with and without yaw misalignment is a topic in itself and considered beyond the scope of this paper (see e.g. (Bromm et al., 2018)).

4) Section 3.1.2: at what downstream distance is the speed deficit measured by the Lidar?

The approximate downstream distance is 2.7D – 100m ≈ 240 m. Note, however, that this value can slightly change based on the instantaneous constellation in the farm, i.e. a yaw angle difference of WT1 and WT2. This is why the consistent probing sector is defined w.r.t. the WT2-based coordinate system – always probing at the fixed distance of [-90m, -110m] upstream of WT2. This is described in the following section 2.3.2.

5) Comments on Fig.12: it is important to notice that the estimator is able to detect wake impingement on both sides of the rotor (left/right). I totally understand that maybe Authors considered it self-evident or trivial, but this is the very first capability that a wake detector must have.

Thank you for highlighting this aspect! You are right that this not a default capability. We mention this aspect in the introduction:

[…] Yet, the observability is limited, as shown e.g. by (Doekemeijer et al., 2020), where the estimator can hardly distinguish which half of the rotor is exposed to a partial wake, especially under uncertain wind direction information. In order to increase the spatial observability of non-uniform turbine inflow, the rotor imbalances - resulting from shear, yaw misalignment or wake impingement - can be encountered […]

**References**

Becker, M., Ritter, B., Doekemeijer, B., Van Der Hoek, D., Konigorski, U., Allaerts, D., & Van Wingerden, J. W. (2022). The revised FLORIDyn model: implementation of heterogeneous flow and the Gaussian wake. *Wind Energy Science*, *7*(6), 2163–2179. https://doi.org/10.5194/wes-7-2163-2022

Bertelè, M., Bottasso, C. L., & Schreiber, J. (2021). Wind inflow observation from load harmonics: Initial steps towards a field validation. *Wind Energy Science*, *6*(3), 759–775. https://doi.org/10.5194/wes-6-759-2021

Bromm, M., Rott, A., Beck, H., Vollmer, L., Steinfeld, G., & Kühn, M. (2018). Field investigation on the influence of yaw misalignment on the propagation of wind turbine wakes. *Wind Energy*, *21*(11), 1011–1028. https://doi.org/10.1002/we.2210

Doekemeijer, B. M., van der Hoek, D., & van Wingerden, J. W. (2020). Closed-loop model-based wind farm control using FLORIS under time-varying inflow conditions. *Renewable Energy*, *156*, 719–730. https://doi.org/10.1016/j.renene.2020.04.007

Dong, L., Lio, A. W. H., & Meng, F. (2021). Wake position tracking using dynamic wake meandering model and rotor loads. *Journal of Renewable and Sustainable Energy*, *13*(2), 023301. https://doi.org/10.1063/5.0032917

Kidambi Sekar, A. P., Hulsman, P., Van Dooren, M. F., & Kühn, M. (2024). Synchronised WindScanner field measurements of the induction zone between two closely spaced wind turbines. *Wind Energy Science*, *9*(7), 1483–1505. https://doi.org/10.5194/wes-9-1483-2024

Lio, W. H., Larsen, G. C., & Thorsen, G. R. (2021). Dynamic wake tracking using a cost-effective LiDAR and Kalman filtering: Design, simulation and full-scale validation. *Renewable Energy*, *172*, 1073–1086. https://doi.org/10.1016/j.renene.2021.03.081

Onnen, D., Larsen, G. C., Lio, W. H., Liew, J. Y., Kühn, M., & Petrović, V. (2022). Dynamic wake tracking based on wind turbine rotor loads and Kalman filtering. *Journal of Physics: Conference Series*, *2265*(2), 022024. https://doi.org/10.1088/1742-6596/2265/2/022024

Ritter, B. (2020). *Nonlinear State Estimation and Noise Adaptive Kalman Filter Design for Wind Turbines* (Issue PhD Thesis).

Soltani, M. N., Knudsen, T., Svenstrup, M., Wisniewski, R., Brath, P., Ortega, R., & Johnson, K. (2013). Estimation of rotor effective wind speed: A comparison. *IEEE Transactions on Control Systems Technology*, *21*(4), 1155–1167. https://doi.org/10.1109/TCST.2013.2260751